JCB | Journal of Cell Biology

## TOOLS

# A proteome-wide yeast degron collection for the dynamic study of protein function

Rosario Valenti[1]*, Yotam David[1]*, Dunya Edilbi[1], Benjamin Dubreuil[1], Angela Boshnakovska[2], Yeynit Asraf[1], Tomer-Meir Salame[3], Ehud Sass[1], Peter Rehling[2,4,5,6], and Maya Schuldiner[1]

Genome-wide collections of yeast strains, known as libraries, revolutionized the way systematic studies are carried out. Specifically, libraries that involve a cellular perturbation, such as the deletion collection, have facilitated key biological discoveries. However, short-term rewiring and long-term accumulation of suppressor mutations often obscure the functional consequences of such perturbations. We present the AID library which supplies "on demand" protein depletion to overcome these limitations. Here, each protein is tagged with a green fluorescent protein (GFP) and an auxin-inducible degron (AID), enabling rapid protein depletion that can be quantified systematically using the GFP element. We characterized the degradation response of all strains and demonstrated its utility by revisiting seminal yeast screens for genes involved in cell cycle progression as well as mitochondrial distribution and morphology. In addition to recapitulating known phenotypes, we also uncovered proteins with previously unrecognized roles in these central processes. Hence, our tool expands our knowledge of cellular biology and physiology by enabling access to phenotypes that are central to cellular physiology and therefore rapidly equilibrated.

## Introduction

The baker's yeast, *Saccharomyces cerevisiae*, stands as a cornerstone in the exploration of cellular biology due to its genetic tractability, cost-effectiveness, and substantial evolutionary conservation with humans (Cohen et al., 2022; Laurent et al., 2020). The advent of genomic collections, also known as libraries, has been a transformative force in yeast studies, enabling genome-wide investigations through arrayed ensembles of strains, each modified at a single gene to facilitate comprehensive studies (Botstein and Fink, 2011).

Notably, the pioneering deletion library (Yeast Knock-Out or YKO) (Giaever et al., 2002; Winzeler et al., 1999), where over 90% of the non-essential genes were deleted in haploid strains, has played a fundamental role in numerous key biological discoveries, accumulating thousands of citations (Giaever and Nislow, 2014). Systematic studies based on the KO library allowed the association of genes with a specific phenotype and from that, cellular roles. For example, the screen for genes required for correct mitochondria distribution and morphology (MDM), carried out by staining the mitochondria of the YKO collection and identifying strains with altered mitochondria, enabled the definition of MDM

strains—many of which were shown to be directly involved in mitochondrial shaping and inheritance (Dimmer et al., 2002). Complementary to the YKO library, the green fluorescent protein (GFP) library (Huh et al., 2003) established a landmark in the systematic visualization of over 60% of the GFP-fused proteome, providing valuable insights into protein abundance and subcellular localization.

However, historically, the construction of such libraries was time-consuming and expensive until the development of the SWAp-Tag (SWAT) libraries (Meurer et al., 2018; Weill et al., 2018; Yofe et al., 2016). The SWAT libraries harbor a placeholder modification in the amino- (N′) or carboxy- (C′) terminus for easy substitution by any genetic sequence of choice such as a modification (tag), promoter, 5′ untranslated region (UTR), and 3′ UTR or selection cassette. Despite these advancements, approximately, a fifth of yeast genes remain uncharacterized or poorly characterized (Cohen et al., 2022; Gaikani et al., 2024). Partially, this can be attributed to limitations in existing tools, such as compensatory mutations arising after gene deletion (Hughes et al., 2000; Teng et al., 2013) that would mask

[1]Department of Molecular Genetics, Weizmann Institute of Science, Rehovot, Israel; [2]Department of Cellular Biochemistry, University Medical Center Göttingen, Göttingen, Germany; [3]Department of Life Sciences Core Facilities, Weizmann Institute of Science, Rehovot, Israel; [4]Max Planck Institute for Biophysical Chemistry, Göttingen, Germany; [5]Fraunhofer Institute for Translational Medicine and Pharmacology, Translational Neuroinflammation 11 and Automated Microscopy, Göttingen, Germany; [6]Cluster of Excellence "Multiscale Bioimaging: From Molecular Machines to Networks of Excitable Cells" 13 (MBExC), University of Göttingen, Göttingen, Germany.

Correspondence to Maya Schuldiner: maya.schuldiner@weizmann.ac.il

*R. Valenti and Y. David are co-first authors.



informative phenotypes, and short-term cellular rewiring, which obscures the correct coupling between perturbation and observed phenotype. For example, transcription of mRNA encoding one paralog is often upregulated upon deletion of the second (Kafri et al., 2005). In other instances, the activation of a transcriptional stress response following the depletion of a single protein can rescue its levels and eliminate the phenotype associated with its loss (Schuldiner et al., 2005).

Addressing these challenges, conditional libraries aimed at altering gene product abundance were developed. For example, tetracycline-controlled gene expression from the TET-off system (Mnaimneh et al., 2004) can repress transcription of the tagged gene upon the addition of doxycycline, a more stable analog of the tetracycline antibiotic. Alternatively, estradiol allows the induction of translation in yeast strains from the YETI library (Yeast with Estradiol strains with Titratable Induction) (Arita et al., 2021). Not surprisingly, essential genes are the main focus of these conditional libraries since, by definition, they were absent from the YKO library. Although useful, these approaches met partial success in eliciting expected phenotypes, potentially due to residual cellular rewiring. Perturbing the translation of a gene or even the levels of its mRNA (Schuldiner et al., 2005) might be too slow to report on an immediate phenotype. One example (Kanemaki et al., 2003) demonstrates that for the essential helicase Mcm4, both a rapid protein destabilization and the repression of transcription were lethal, but the different rates of protein depletion in each method resulted in cells being arrested in different stages of the cell cycle. Indeed, rapid protein destabilization, as achieved with thermosensitive (TS) strains (Kofoed et al., 2015; Li et al., 2011), has historically been used to probe the role of essential components and was instrumental in the discovery of the cell division cycle (CDC) genes (Hartwell et al., 1973; Hartwell et al., 1970). CDC mutants in TS strains grown in their restrictive temperature perished while displaying a synchronized morphology, setting the basis for the discovery of the cyclins and cyclin-dependent kinases currently known for governing the cell cycle progression. This example not only highlights the power of yeast genetics but also the complementarity offered by different strategies and the importance of having tools for rapid, on demand, protein depletion. More recent advancements, such as the generation of a collection of over 750 strains carrying an auxin-inducible degron (AID) (Snyder et al., 2019) showcase the potential of systems enabling protein degradation induced by the addition of a small ligand.

Motivated by the notion that new tools drive biological advancements, we opted for a degron-based conditional protein depletion strategy. We chose the improved auxin-inducible degron system (AID2) (Yesbolatova et al., 2020) as a rapid, reversible, and flexible system that induces the depletion of the targeted protein upon the addition of a chemically modified version of auxin, the small molecule 5-phenyl 1H-indole-3-acetic acid (5-Ph-IAA). This system enjoys baseline residual degradation in the absence of external stimulus thus making it much less taxing on cells and more reactive to experimental conditions. Leveraging the C′ SWAT library, we extended our approach to

the entire proteome allowing us to screen for immediate phenotypes following degradation of both non-essential and essential proteins. In our design, we also incorporated a GFP tag for simultaneous reporting on protein abundance, localization, and the effects of protein deletion—a conceptual fusion of the GFP library with the YKO.

We validated our C′ AID-GFP library by revisiting seminal yeast studies, uncovering additional essential proteins under different growth conditions, recapitulating and extending the MDM screen, and systematically detecting strains with synchronized morphologies in a reapproach of the CDC screen. Successful reproduction of these studies and identification of new gene functions attests to the complementarity of our approach within the expansive toolbox of yeast genetics. Our library will be freely distributed hoping to fuel a wave of discoveries on yet uncharacterized yeast protein functions.

## Results

### Establishing a system for fast and tractable protein depletion based on the AID2 system

To induce physiologically relevant and informative perturbations while minimizing cellular rewiring, our goal was to establish a rapid and efficient protein depletion system. To this aim, we adapted the improved AID2, a versatile tool for dynamic, on demand depletion of endogenous proteins in yeast cells (Yesbolatova et al., 2020). This system is superior to previous ones in that background degradation by auxin-like cellular molecules is dramatically diminished, minimizing background degradation in the absence of the inducer. The system is also more specific, requiring a low dose of inducer that presents no toxicity to the cells and that can be applied under diverse conditions without generating inherent stress to the cells.

Our implementation of the AID2 system involves fusing the small AID* tag (amino acids 71–114 from AtIAA17, Uniprot ID: P93830, from hereon termed simply AID) (Morawska and Ulrich, 2013) followed by the enhanced eGFP (Zacharias et al., 2002), for visualization, to the C′ of a protein on the background of constitutive expression of the adaptor OsTIR1(F74G). These modifications did not have a big impact on yeast growth (Fig. S1 A). The system is induced by the addition of the modified auxin, 5-Ph-IAA, which leads to the recognition of the AID tag on the protein of interest by the TIR1 adaptor protein. TIR1 binding to the AID recruits the ubiquitination machinery (Skp1-Cul1 E3 ligases [Yesbolatova et al., 2020; Yu et al., 2015]) and leads to proteasomal degradation of the AID-tagged protein (Fig. 1 A). We did not observe any growth deffect from the 5-Ph-IAA treatment, neither in the strains containing the OsTIR1(F74G) nor their parental strains (Fig. S1 B).

To ensure that GFP fluorescence accurately reports on protein degradation rates, we monitored the degradation of various proteins using both GFP fluorescence imaging (Fig. 1 B) as well as western blot analysis (Fig. 1 C). We confirmed the correspondence between protein levels and GFP fluorescence (Fig. 1 D), irrespective of the protein's half-life, and also that free GFP does not accumulate during degradation. Since GFP imaging

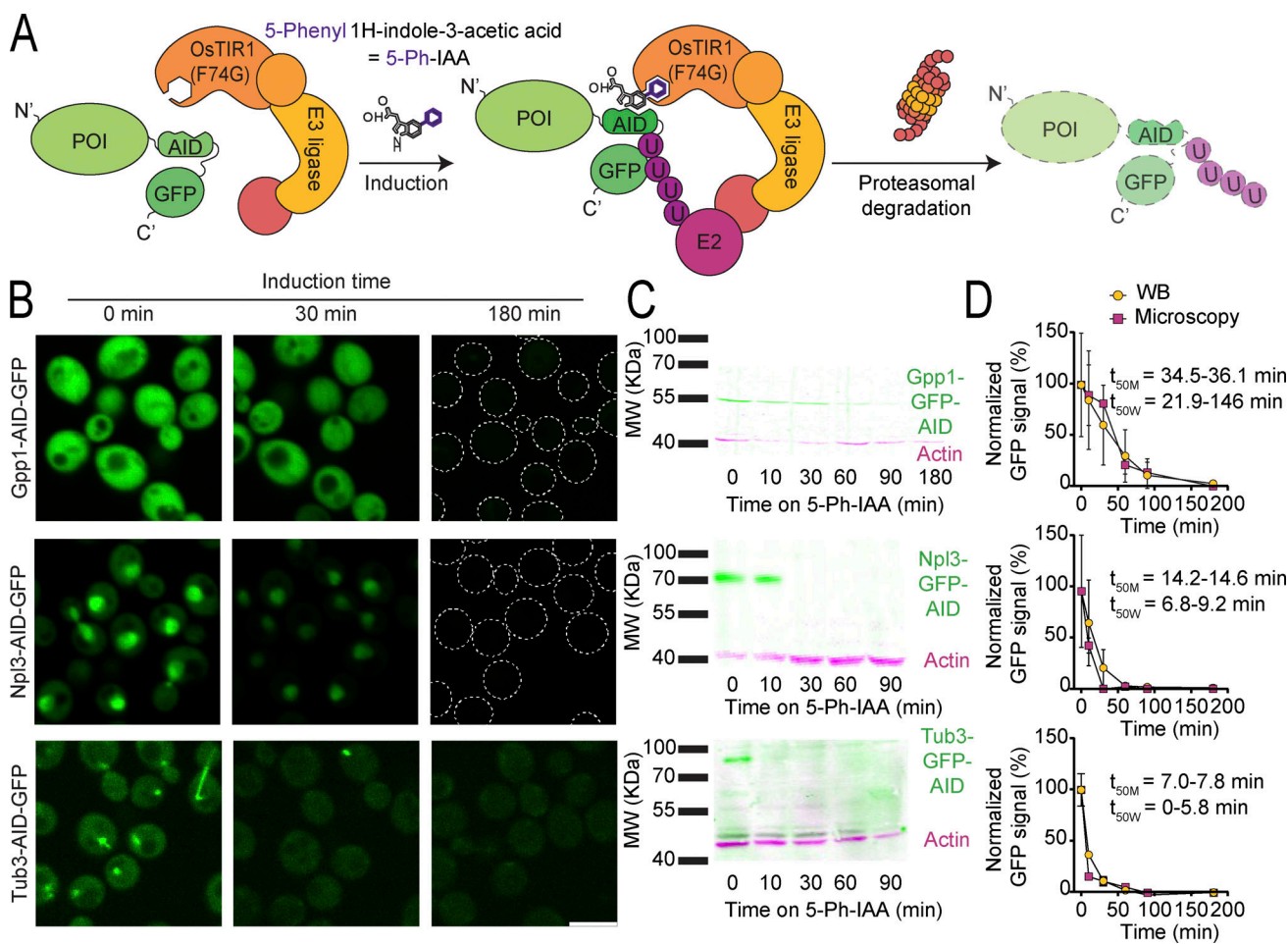

Figure 1. **The auxin-inducible degron (AID) system allows "on demand" depletion of the proteome. (A)** Schematic representation of the AID system, where a protein of interest (POI) is fused to the AID2 tag and a GFP for visualization. In the same cells, the OsTIR1(F74G) adaptor protein is constitutively expressed. Upon induction by the addition of the modified auxin, 5-Ph-IAA, the construct gets ubiquitinated and degraded via the proteasome. **(B)** Fluorescent images of selected proteins tagged with the AID system after 0, 30, and 180 min of induction. All the proteins are depleted either partially or below detection levels upon induction. For Gpp1 and Npl3 at 180 min, the outline of the cells is depicted with a dotted line. Scale bar: 5 μm. **(C)** Western blots (WB) of the same strains shown in panel B. Immunoblotting was performed against GFP (green) and actin (magenta) as a loading control. The GFP-tagged proteins are depleted partially or below detection levels upon induction. Only one band is detected, at the expected molecular weight in each case, indicating no free GFP or intermediate degradation stages are generated. **(D)** Graphs comparing the degradation curves for each protein from panel B, based either on the fluorescence microscopy images (yellow circles) or on the western blots from panel C (pink squares), normalized to time 0. Both methods produced comparable curves, despite the half-life ($t_{50M}$ for microscopy and $t_{50W}$ for WB) of the proteins spanning a wide range of time. Three technical replicates were used in each case. A one-phase decay curve was fitted to the data to estimate the half-lives of each protein and the 95% confidence interval of the $t_{50}$ is displayed. Source data are available for this figure: SourceData F1.

presented higher throughput and additional insights into the subcellular localization of the remaining protein, we turned to measuring GFP fluorescence changes as a reliable indicator of protein degradation.

### Creating a proteome-wide collection of depletable proteins and characterizing responsiveness at a proteome level

To enable the comprehensive characterization of yeast phenotypes following "on demand" elimination, we extended the modified AID2 system to a whole-proteome yeast library using the recently developed SWAT approach for library generation (Meurer et al., 2018; Weill et al., 2018; Yofe et al., 2016) (Fig. 2 A). Following the "SWATting" procedure, we had 92% of the strains present in the new library (survival rate [Fig. S1 C]). We

estimated 94% of "SWATting" efficiency, considering the fraction of strains with effectively detected fluorescence signals among the 500 most abundant proteins (Ho et al., 2018) (Fig. S1 D).

We also sequenced the library to validate "in frame" insertion of the tag, which was confirmed for 5,058 strains (Fig. S1, E and F; and Table S1). Hence, we successfully created the C' AID-GFP library comprising 5,170 arrayed strains, each constitutively expressing the TIR1 adaptor protein and containing a unique gene fused with the AID tag and GFP at their C' terminus.

Upon imaging the entire collection, ~60% of the strains exhibited a fluorescent signal, consistent with reports suggesting that nearly 2,000 proteins are either not expressed at all or are in very low abundance under standard laboratory

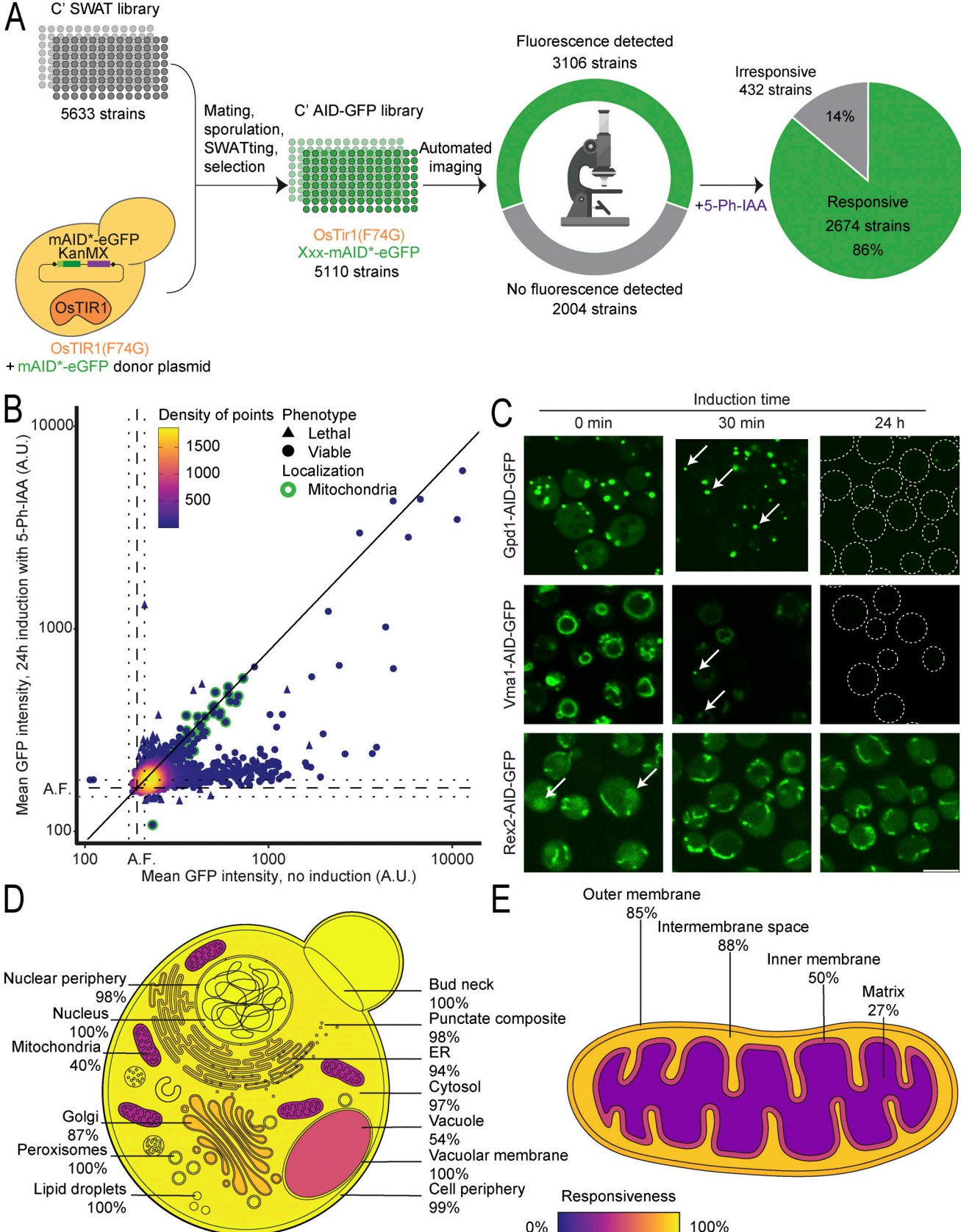

Figure 2. **The C' AID-GFP library has robust responsiveness across cellular compartments. (A)** Summary schematics of the expansion of the AID system to a whole genome collection. The original C' SWAT library was used to generate the C' AID-GFP library using an automated process of mating, selection, sporulation, and SWATing. In the final library, every strain expresses the OsTIR1(F74G) protein, and a different gene is fused to the AID-GFP tag. Fluorescence

was detected in 60% of the strains under standard laboratory conditions (log-phase cells in rich media). Of those, 86% responded to the induction with a measurable protein depletion. **(B)** Dot plot showing the mean GFP fluorescence intensity per cell for each strain before or after 24 h of induction time. The background autofluorescence noted A.F. calculated as the mean fluorescence of control strains is marked in dotted lines with a confidence interval of two standard deviations in each direction. The diagonal line indicates the expected location for irresponsive strains. Many strains die after induction (triangles), making their fluorescence signal not informative, while most of the irresponsive strains encode for mitochondrial proteins (green edge). **(C)** Fluorescent images of selected proteins were imaged after no (0 min), short (30 min), or long (24 h) periods of induction. In the case of dually localized proteins, induction depletes one of the subpopulations with faster kinetics. Gdp1 remains visible in peroxisomes, Vma1 in endosomes and Rex2 in mitochondria while their respective cytosolic, vacuolar, or nuclear subpopulations are decreased. In cases where the signal is too dim, the outline of the cells is depicted with a dotted line. Scale bar: 5 μm. **(D and E)** Schematics of a yeast cell in panel D and a mitochondrion in panel E displaying the responsiveness per subcellular compartment. Proteins in membrane-bound organelles are more protected from degradation but the responsiveness across subcellular compartments is still high.

---

conditions (log-phase growth in rich media) (Breker et al., 2013; Ghaemmaghami et al., 2003; Huh et al., 2003). Inducing the AID system led to the reduction of fluorescence levels, either partially or below detection limit, for nearly 90% of the strains evaluated under these conditions (Fig. 2 A). This high responsiveness underscores the broad applicability of our system for effectively knocking down proteins in diverse cellular contexts.

Since we observed variations in the depletion dynamics among individual proteins (Fig. 1, B–D), we systematically analyzed the response of each strain to short (30 min) and long (24 h) periods of induction. We imaged the AID strains by high-throughput fluorescence microscopy before and after induction, with actively dividing cells in rich media. The resulting images were manually inspected, and combining the manual examination results with a comprehensive quantitative analysis, we determined responsive strains with a significant reduction of their GFP signal after induction. Within our collection, we identified proteins that exhibited rapid depletion, such as Lia1, while others required an extended induction time but were depleted below detection limits after 24 h of induction, exemplified by Mrh1. Concurrently, some proteins, like Tim44, displayed negligible responsivity to the AID system even after a long induction time of 24 h (Fig. S2 A).

A short induction time was sufficient to measure changes for 1,642 strains (Fig. S2 B), although a clearer response emerged after 24 h of induction, with 2,674 strains, representing 86% of the visible proteome, displaying responsivity (Fig. 2 B). Notably, after this extended induction, we observed the demise of certain strains (Fig. 2 B, triangles). This induction-provoked lethality implies the depletion of an essential protein in those strains, confirming that the AID-tagged protein was crucial for cell viability. If we consider these strains too, then overall, our library has 2,713 clearly responsive strains.

Interestingly, we noticed that over 100 dually localized proteins exhibited distinct depletion kinetics for each subcellular population (Fig. 2 C). Exploiting these differential degradation rates provided enhanced visualization of the eclipsed subpopulations, particularly notable after short induction periods (Fig. 2 C). This enables visualizing localizations for several proteins previously thought to be solely cytosolic.

Additionally, we expected proteins behind cellular membranes to face challenges accessing the ubiquitination targeting and proteasomal degradation machinery upon induction. Indeed, we observed many irresponsive strains with their tagged protein localized to the mitochondria (Fig. 2 B, green edge) or to the lumen of other organelles, in line with our expectations.

Despite this limitation, the responsiveness per cellular compartment was robust, ranging from 40% to 100% (Fig. 2 D). Even within mitochondria, responsiveness correlates with the suborganellar compartment, with the mitochondrial matrix proving to be the most challenging subcompartment to target (Fig. 2 E). We attributed the weaker response to the time spent by proteins dwelling in the cytosol before translocating into mitochondria, though milder depletion could also reflect longer protein half-lives (Bomba-Warczak and Savas, 2022). Proteins lingering longer in the cytosol are more susceptible to the clearance of newly synthesized proteins. The remaining protected protein pool may be diluted by cellular divisions or depleted through natural turnover. Segregating mitochondrial proteins into co- and posttranslationally translocated groups (Williams et al., 2014) (Fig. S2 C), supports our hypothesis about higher degradation with increased residency in the cytosol. Notably, posttranslationally translocated proteins are 2.2 times more likely to be responsive to induction of degradation. A similar trend is observed with the C′ orientation of membrane proteins (Weill et al., 2019), where exposure of the AID element to the cytosol leads to more efficient degradation, as exemplified for the endoplasmic reticulum (ER) (Fig. S2 D).

Overall, even when protein abundance correlated with quantified fluorescence as expected (Fig. S2 E), the responsiveness of the fluorescent fraction remained quite steady (Fig. S2 F). Between the first and last decile of abundance, responsiveness ranged from 82 to 92% ruling out the dependence of induced degradation on protein abundance. This seems to indicate that the ubiquitin–proteasome system is not a limiting factor, and therefore responsiveness is not conditioned by the abundance of the protein.

## Application of the C′ AID-GFP library to the study of essential proteins

To assess the utility of our library, we revisited seminal yeast studies focusing on central cellular processes and examined our collection's ability to recapitulate known phenotypes as well as its potential to uncover genes not previously known to affect these processes.

Encouraged by the observation of induction-provoked lethality in certain strains (Fig. 2 B), our initial investigation concentrated on determining protein essentiality. To this end, we cultured our arrayed yeast collection on three types of solid media that represent different yeast growth physiologies: rich media, minimal media, or a non-fermentable carbon source, with or without 5-Ph-IAA, to induce the degradation of the

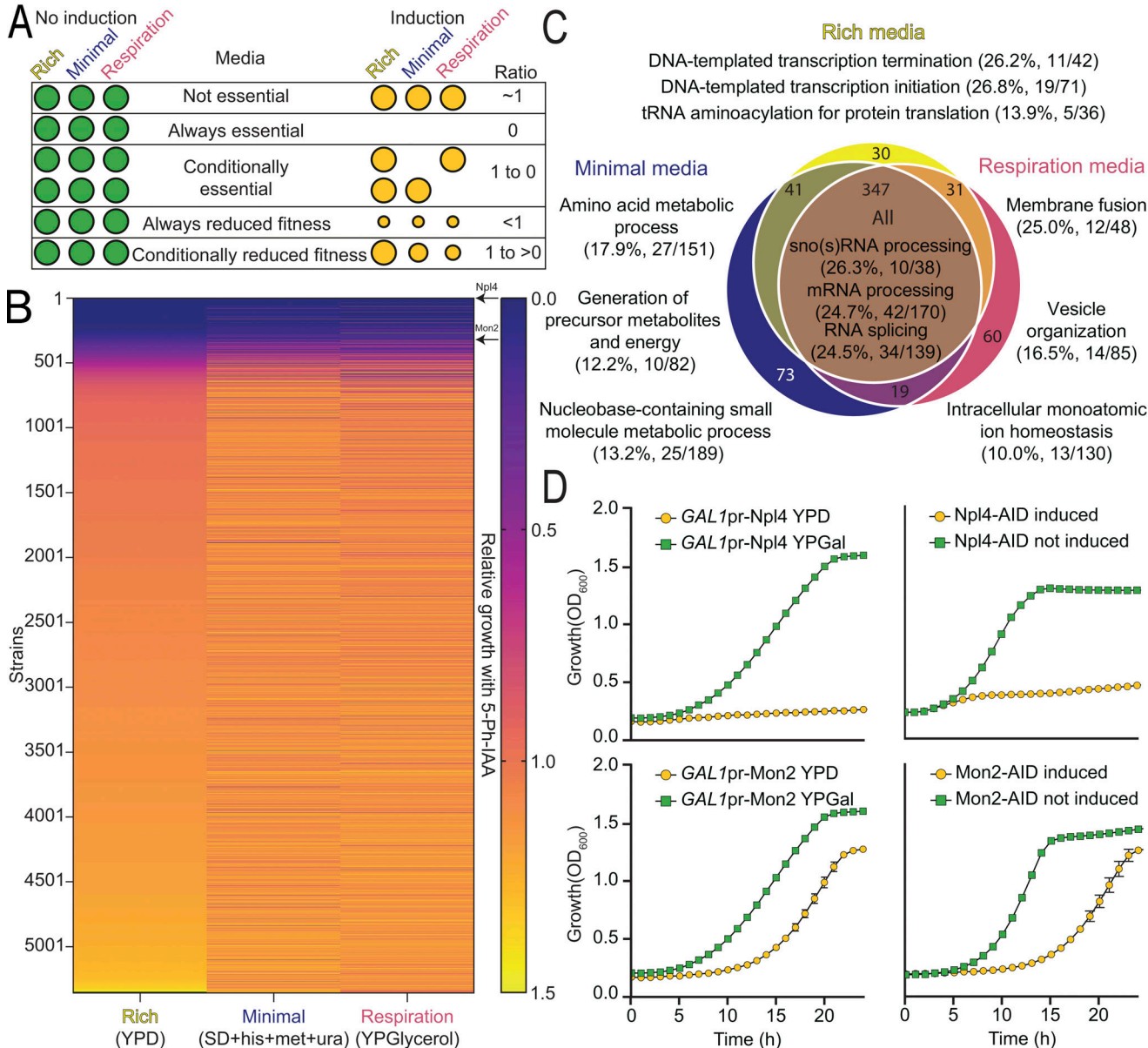

Figure 3. **The C' AID-GFP library can be applied to the study of protein essentiality. (A)** Schematics representing our pipeline to test protein essentiality in different media. The arrayed yeast collection can be grown in different types of solid media with and without induction of the AID system. The ratio of the colony size, or relative growth, is indicative of whether the protein depleted is important for growth rate or viability. **(B)** Heatmap showing the relative growth of every strain in the collection in three different media. The vast majority of the strains (89%) display no severe growth defect upon induction of degradation, except for 601 strains. The results are very similar for all the media tested, with specific strains displaying altered growth in one or more media. Strains were grown in duplicates and for two consecutive plate replications. **(C)** Venn diagram summarizing the results from panel B for all the strains with a growth defect bigger than 50%. Most of these strains (347) displayed a severe growth defect in every media. The top three GO slim terms for biological processes associated with each group of strains are displayed, together with the percentage of the GO slim term covered in the group. A clear relationship between the media analyzed and the processes affected can be appreciated. **(D)** Growth curves of a selected example for a protein that is essential, as Npl4, or cause a fitness loss, as Mon2, upon depletion (orange series). Both our system and a *GAL1*-promotor swap reproduce the results seen in panel B. The average of three repeats with their standard deviation is shown.

tagged proteins. Colony sizes were measured after growth in induced versus uninduced conditions. The ratios of colony sizes scored the extent of the relative growth defect due to induced protein degradation among the strains (Fig. 3 A). Most strains grew similarly across all three media tested, including 347 strains presenting severe defects (characterized by a relative growth score below 0.5) (Fig. 3 B and Table S1). In addition, 254

strains manifested growth defects in only one or two of the tested media (Fig. 3 C).

To compare the results obtained in each media, we calculated the enrichment of Gene Ontology (GO) slim terms related to biological processes (Ashburner et al., 2000; Gene Ontology Consortium et al., 2023). GO slim, the cut-down version of GO terms, provides a broad overview of biological processes

associated with a particular set of genes. We grouped strains with severe growth defects for each media (Fig. 3 C). We listed the top three associated GO slim terms, along with the percentage of each term recovered in their corresponding group of strains. As anticipated, fundamental processes, such as those related to RNA processing, emerged as top hits for strains affected in all tested media. Processes essential for minimal media, relating to the biosynthesis of amino acids or metabolites, were also prominent in their corresponding group, aligning with the absence of external nutrient provision.

Analysis of all strains displaying severe defects also uncovered new candidate essential genes that were previously missed. The first systematic classification of protein essentiality in yeast resulted from the creation of the YKO library, where the definition of an essential protein came from the unattainable gene deletion of a haploid strain despite being created as a heterozygous diploid (Giaever et al., 2002). Examining only our responsive strains, the list of strains with severe growth defects (relative growth threshold <0.5) recovered 69% of those originally defined essential genes (Fig. S3 A, left). To favor the retrieval of *bona fide* unidentified essential genes over slow-growing strains, we used a stringent threshold of <10% relative growth. Among the 281 strains selected, only 11 had not been identified as essential in the YKO (Fig. S3 A, right), even though five were reported as essential genes in the literature from low-throughput experiments, supporting our findings (DeHoratius and Silver, 1996; Hampton et al., 1996; Hoffman et al., 2016; Kastenmayer et al., 2006; Li et al., 2005; Piłsyk et al., 2020; Stirling et al., 2011).

To validate the reproducibility of our high throughput screen, we selected two strains and repeated the assay in liquid media using both the AID induction and an orthogonal approach of changing their promoter to a repressible one. Replacing the endogenous promoter with the galactose inducible/glucose repressible, *GAL1*, promoter, we depleted desired proteins by growing cells on media with glucose overnight. Both methods in liquid growth assays recapitulate the results obtained from the high-throughput approach (Fig. 3 D). In the future, it would be worthwhile to find essential genes under numerous diverse conditions, which is now possible since the AID approach is robust in a wide array of conditions (for examples see Fig. 3 C). Furthermore, many conditional depletion systems have been shown to be more efficient with highly expressed genes (Arita et al., 2021). We noted that our system follows the opposite trend (Fig. 3 B), making it highly complementary to other approaches.

**The C′ AID-GFP library uncovers additional proteins required for normal mitochondrial distribution and morphology**
Given that mitochondria imposed a tough challenge for our approach, we sought to assess specifically the capability of the collection to yield meaningful insights into mitochondrial cell biology. To address this, we turned to a screen that systematically defined genes essential for correct mitochondrial distribution and morphology (MDM) using the deletion collection for non-essential genes (Dimmer et al., 2002) or the TET-off repressible collection for essential ones (Altmann and Westermann,

2005). Mimicking their approach, we employed a mitochondrial dye to observe mitochondria morphology in strains from our collection, following 24 h of induction of the degron system.

In line with prior discoveries, we successfully recapitulated known MDM phenotypes (Fig. S4 A). Within our responsive strains, we rediscovered 60% of the proteins previously documented to induce mitochondria morphology defects when depleted. Differences between the studies may stem from our inability to deplete certain mitochondrial proteins to the extent required for the phenotype to arise, or the result of changes that occur only after a strain has lost its mitochondrial DNA. Such a process typically takes longer than 24 h and would therefore only be present in deletion strains. Moreover, we identified over 220 previously undescribed candidates where mitochondrial morphology appeared altered, as revealed in a double-blind manual analysis and quantification of the images (Table S1). Examples of these newly described MDM-like strains (Fig. 4 A) are proteins from pathways like lipid biogenesis, RNA processing, cytoskeleton organization, and many more. To relate the morphological changes to physiological perturbations, we measured the oxygen consumption rate from respiration for selected strains (Fig. 4 B and Table S6). We did not observe significant changes in any of the controls (uninduced AID strains or the control strain either before or after induction). However, significant changes (one-way ANOVA, FDR 0.05) were observed after induction for those strains with mitochondrial morphology defects (Fig. 4 B). These physiological changes underscore the diverse biological processes regulating mitochondrial function and emphasize the invaluable analytical perspective our library provides in studying mitochondria.

Also in accordance with previous studies (Altmann and Westermann, 2005; Zung et al., 2024, *Preprint*), systematic evaluation of mitochondria images after 5-Ph-IAA induction revealed an increased mitochondrial fluorescence signal following the depletion of most ergosterol biosynthesis-related proteins (Fig. 4 C). Since this is an essential pathway, we repeated the experiment but with only 4 h of induction of the AID system to prevent cell death from interfering with our observations. Even after this short depletion time, the mitochondrial morphology and the fluorescent signal's intensity of most strains were altered (Fig. 4 D and Fig. S4 B), irrespective of whether they participated in the pre- or post-squalene module of the pathway (Fig. S4 B). Unsurprisingly, exceptions are the strains where Hmg1 or Hmg2 were tagged, in which the depletion of one paralog can be compensated by the action of the other (Basson et al., 1986); or Erg11, where the depletion is slower (Fig. S4 B). Notably, the morphological modifications were also elicited by fluconazole and terbinafine, two drugs that inhibit the ergosterol biosynthesis pathway (Fig. S4 C) (Zung et al., 2024, *Preprint*). To assess whether the increased fluorescent staining stems from a physiological change in mitochondrial respiration, we assayed their respiration-related oxygen consumption rate (Fig. 4 B and Fig. S4 D). This respiration-related oxygen consumption is the difference between basal oxygen consumption and non-respiration oxygen consumption (measured in the presence of the respiration inhibitor antimycin A to account for the oxygen consumed in additional cellular processes, such as

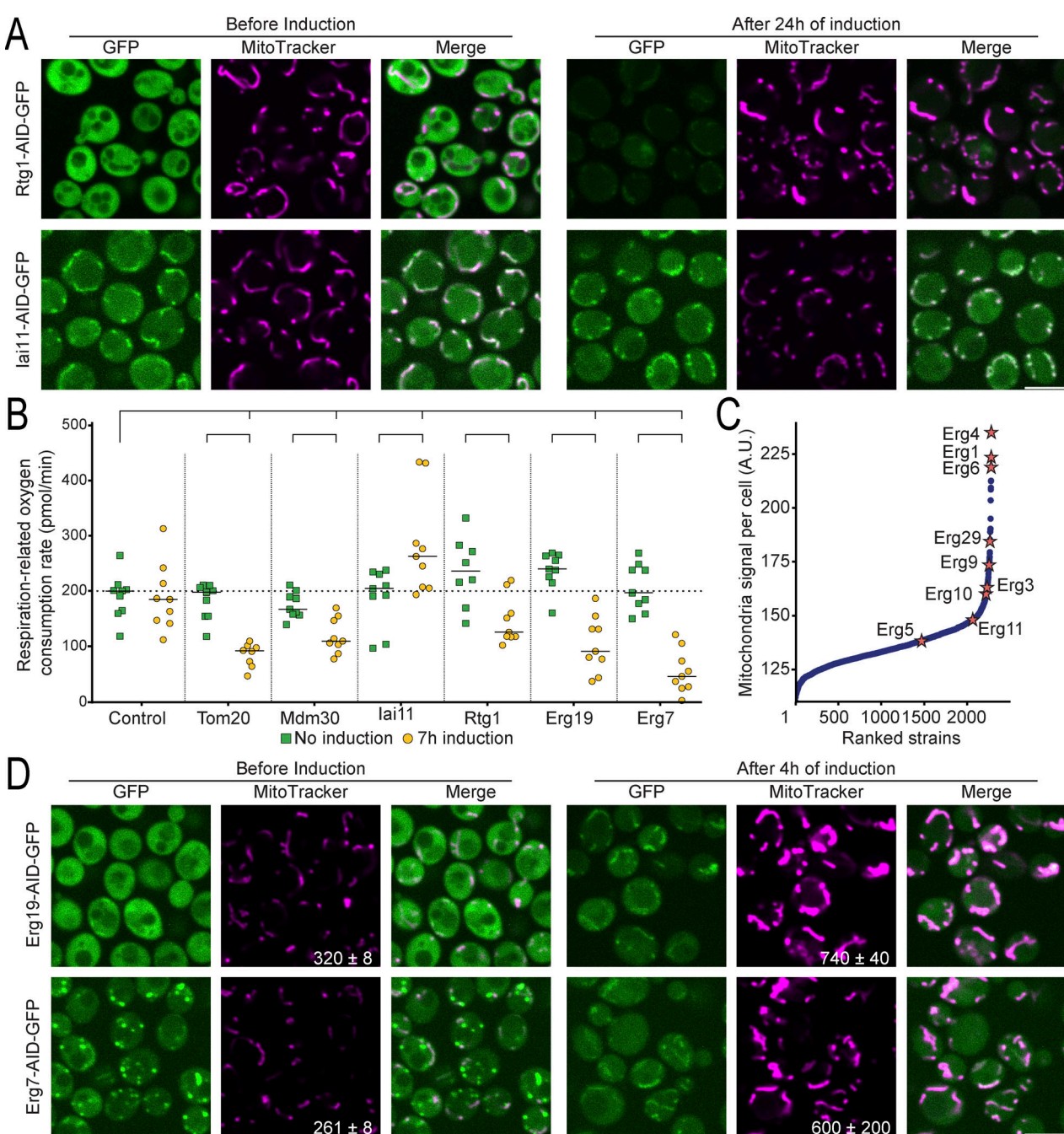

Figure 4. **New mitochondrial morphology and distribution phenotypes appear after protein depletion. (A)** Fluorescent images of strains before and after induction of the degron system for 24 h, for previously undescribed proteins required for correct mitochondria morphology and distribution. After depletion of the tagged proteins (green), the mitochondria look aberrant (magenta). **(B)** Bar graph showing the oxygen consumption rate derived from respiration for control and selected strains, before and after 7 h of induction with 5-Ph-IAA. Significant changes for each group with the control and for each induced versus their corresponding uninduced group are marked (one-way ANOVA, FDR 0.05). **(C)** Scatter plot showing the MitoTracker signal intensity per cell after 24 h of induction versus strain rank. Many proteins belonging to the ergosterol biosynthesis pathway (stars) display elevated mitochondrial signal intensity (which could either be changes in membrane potential or in shape) compared to the median. **(D)** Fluorescent images of selected strains whose depleted protein participates in the ergosterol biosynthesis pathway, without or with 4 h of induction. The mitochondrial morphology changes and the MitoTracker signal intensity (quantification of mean MitoTracker signal per cell ± SD in the MitoTracker images) increases (magenta) upon protein depletion (green). For all images, the scale bar is 5 μm.

ergosterol biosynthesis itself). We found that depleting proteins related to the ergosterol biosynthesis pathway caused an increase in oxygen consumption not related to respiration (Fig. S4 E), as expected and in accordance with previous reports (Zung et al., 2024, *Preprint*). Importantly, they displayed reduced respiration (Fig. 4 B, Fig. S4 D, and Table S6) supporting the ability of our assay to uncover regulators of mitochondrial respiration. Collectively our data highlights an important role for sterols in

maintaining mitochondrial shape and function, and moreover, they suggest that additional factors influencing mitochondrial biogenesis await discovery.

**Revisiting the cell division cycle screen using the C′ AID-GFP library suggests additional genes required for cell division**

Finally, we revisited the CDC screen (Hartwell et al., 1970), which uncovered proteins essential for normal cell division and cell cycle progression. Since the original screens required the creation of TS strains via mutagenesis, while the C′ AID-GFP library encompasses most of the proteome, we speculated there might be proteins with CDC-like features still unreported. From our long-term induction screen, it became strikingly apparent that many strains, not only those of previously known CDCs, exhibited altered synchronized morphology (Fig. 5, A and B). After 24 h, by visual examination, we systematically identified all strains arrested with a synchronized morphology (Table S1). Among those strains, we retrieved two well-characterized complex members of one of the original CDC mutants, Cdc48–Npl4 and Ufd1. Cdc48 has been thoroughly characterized as an AAA ATPase that governs protein degradation and its depletion causes the cells to arrest in the G2 phase (Fig. S5 A). Despite not previously being characterized as such, we found that Npl4 displays the same CDC characteristics as its interactor Cdc48 (Fig. S5 B).

Encouraged by the sensitivity of our screen, we decided to follow up on some of the strains not previously annotated as CDCs. In addition to the synchronized altered morphology (Fig. 5 B), the induction of degradation led to growth defects (Fig. 5 C) and altered flow cytometry profiles (Fig. 5 D). Notably, we observed cases of cell cycle arrest in various phases, some with literature support: G1 (e.g., Rio1 [Angermayr et al., 2002; Iacovella et al., 2015]), G2 (e.g., Npl4 [Hsieh and Chen, 2011] and Sts1), as well as issues in S phase (e.g., Pol12 [Yu et al., 2006] and Sld3 [Kamimura et al., 2001]). We also documented instances where cells continued to duplicate their DNA without undergoing cellular division (e.g., Chs2 [Schmidt et al., 2002] and Ndd1 [Loy et al., 1999]). These results collectively pinpoint the utility of the C′ AID-GFP library as a valuable tool for investigating the cell cycle and cell division processes.

## Discussion

Cellular perturbation has been an invaluable approach to uncovering the role of many components in the cell. Inducible systems, in particular, enable the study of dynamic processes and essential components. In this study, we (see also a similar library created in parallel [Gameiro et al., 2024, *Preprint*]), have created the C′ AID-GFP library, a novel proteome-wide collection designed for on demand protein depletion to systematically explore cellular processes. Leveraging the versatility of the AID2 system (Yesbolatova et al., 2020) and the efficiency of SWAT libraries (Meurer et al., 2018; Weill et al., 2018; Yofe et al., 2016), we developed a comprehensive tool for investigating protein function and cellular responses.

We demonstrated the robustness of our approach, with nearly 90% of the expressed proteins responding to induction of the degron (Fig. 2 A). Moreover, we displayed its applicability by revisiting seminal yeast screens, recapitulating their results, and extending them. We found our system to be highly complementary with other collections that focus on essential genes, such as the YETI-E (Arita et al., 2021) and the TET-off (Mnaimneh et al., 2004) (Fig. S3 B), as it effectively evokes the essentiality of lowly expressed proteins. We observed that for some abundant proteins, the depletion was incomplete. For some essential proteins, this might have limited our capacity to evoke their lethality. In the future, the study of such proteins could be improved by incorporating a transcriptional repression strategy thus allowing tighter control of the protein levels. Importantly, our library represents the majority of yeast genes, also allowing us to uncover novel essential genes and new phenotypes for non-essential ones.

Indeed, among our results, we identified essential proteins absent or misannotated in the YKO library (Fig. S3 A). Reports in the scientific literature positively support our results with high-confidence demonstrating how the C′ AID-GFP library is a well-suited tool to elicit induction-provoked lethality. In our list of high-confidence new essential proteins, we encounter *YIL102C-A*. Upon closer inspection of this protein, labeled as uncharacterized by the *Saccharomyces* Genome Database (SGD) (Engel et al., 2022; Wong et al., 2023), it appears to have been already characterized as a regulatory subunit of dolichyl phosphate mannose (DPM) synthase (Piłsyk et al., 2020). To this end, we propose *YIL102C-A* to be renamed as *DPM2*, as it's human counterpart (Maeda et al., 1998).

To ensure widespread accessibility to the scientific community, we openly distribute this library and have compiled our results into yDIMMER (yeast Degron Induced Mitigation for Molecular Exploration of Responses), a user-friendly database: https://www.weizmann.ac.il/ydimmer/ (Fig. S6). Visualization of any strain present in our library, before or after either short or long induction times, is accessible through this platform. Additionally, we provide access to quantitative information about each strain's relative growth in different media, as well as their mitochondrial morphology. The website supports single gene searches, bulk download of information, and filtering by specific properties (e.g., strains with growth defects in respiration media).

In conclusion, our study introduces a powerful tool for functional genomics research, opening up new possibilities for investigating cellular processes. The C′ AID-GFP library, with its ability to systematically deplete proteins and assess cellular responses, promises to accelerate discoveries in yeast biology and beyond.

## Materials and methods

### Yeast strains and plasmids

All yeast strains used in this study are listed in Table S2. Strains were constructed either by transformation, using the lithium acetate-based transformation protocol (Daniel Gietz and Woods, 2002), or by SWATting (Meurer et al., 2018; Weill et al., 2018; Yofe et al., 2016). All plasmids used are listed in Table S3. The

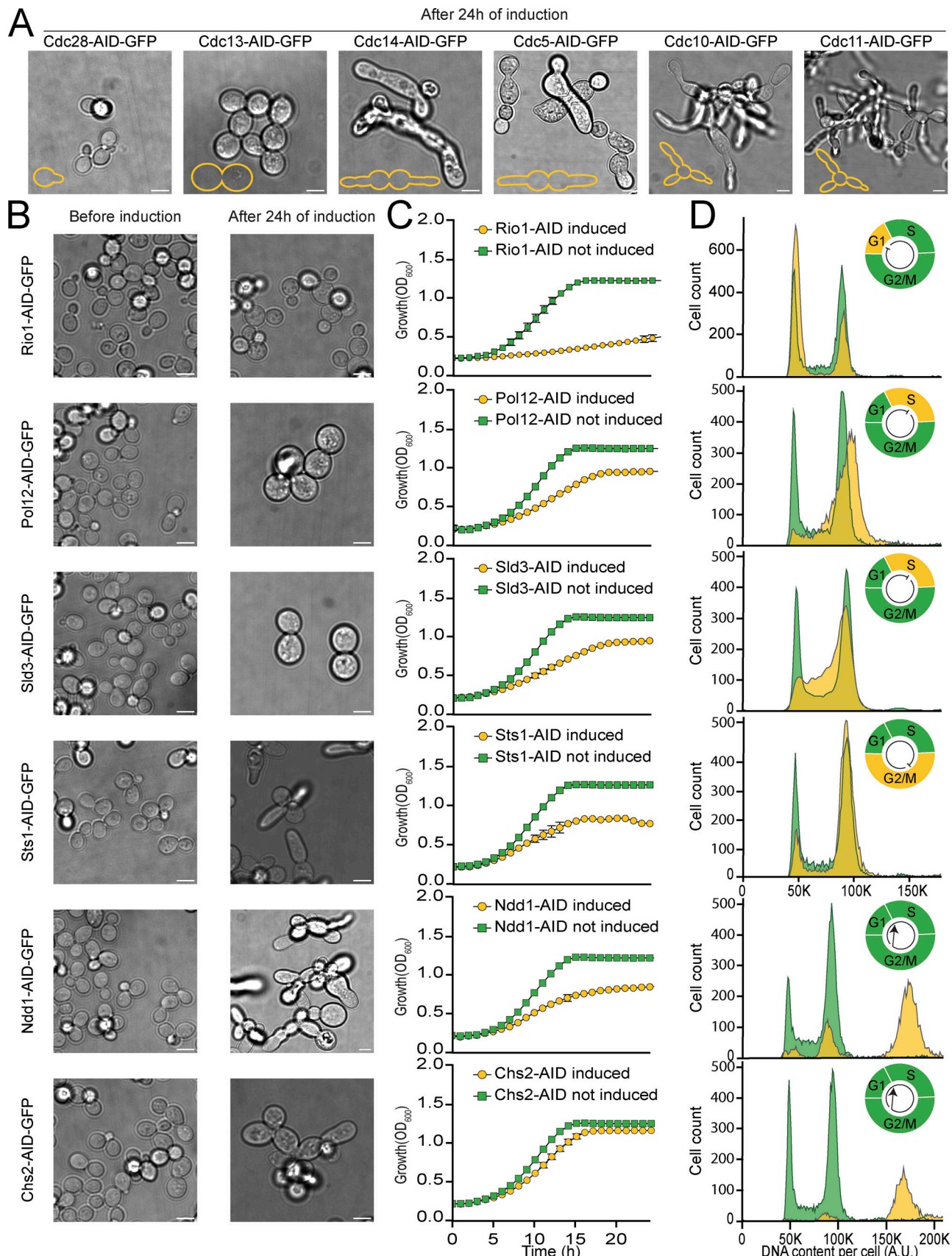

Figure 5. **The C'AID-GFP library facilitates the study of CDC genes. (A)** Brightfield microscopy images of strains previously assigned as CDCs, after induction of the degron system for 24 h. A small schematic, recreating the original cartoons (Hartwell et al., 1973) is depicted in the lower left corner

of each image. Scale bar: 5 µm. **(B)** Brightfield microscopy images of strains here suggested as CDCs, before and after induction of the degron system for 24 h. In every case, a distinct and synchronized altered morphology can be observed after protein depletion. Scale bar: 5 µm. **(C)** Growth curves for strains in panel B, with induction of the degron (orange curve) or not (green curve). Growth defects or lethality are seen upon depletion of the indicated tagged protein. An average of three repeats with their standard deviation is shown. **(D)** Flow cytometry measurements for strains in panel B, with or without a 3 h induction. Without induction (green), a typical histogram for cycling cells is appreciated in each case, while different phenotypes point to issues in the cell division cycle in each protein-depleted case (orange). A schematic of the cell cycle (upper right corner of each panel) shows which phase is affected (orange blocks and arrows).

donor plasmid was constructed from a plasmid kindly provided by Nir Friedsman lab with the original SWAT donor plasmid. Primers were designed with the Primers-4-Yeast web tool (https://www.weizmann.ac.il/Primers-4-Yeast/ [Yofe and Schuldiner, 2014]), Table S4, includes all the check primers to verify the library clones used in the figures. A SWAT donor strain was constructed by integration of the TIR1(F74G) adaptor protein into the original SWAT donor strain (yMS2085; [Weill et al., 2018]) at the *BPT1* locus to create (yMS6680). This strain displays a small growth defect compared with its parental strain (yMS2085). The resulting strain was transformed with a SWAT donor plasmid encoding the C′ AID-GFP tag and crossed with the C′ SWAT library ([Meurer et al., 2018]).

**Yeast growth**
Yeast cells were grown on solid media containing 2.2% [wt/vol] agar (#AGA0X; Formedium) or liquid media, at 30°C. Antibiotics nourseothricin (NAT, WERNER BioAgents "ClonNat" #5.XXX.000) at 0.2 g/liter and geneticin (G418, #G418X; Formedium) at 0.5 g/liter were used for library maintenance and overnight precultures. Unless stated otherwise, the media used was YPD (YPD broth 5% [wt/vol] Formedium #CCM02XX: 2% [wt/vol] peptone, 1% [wt/vol] yeast extract, and 2% [wt/vol] glucose). For microscopy, the media used was synthetic minimal media (SD; 0.67% [wt/vol] yeast nitrogen base [YNB] without amino acids and with ammonium sulfate [#CYN04XX; Formedium], 2% [wt/vol] glucose), supplemented amino acid OMM mix ([Hanscho et al., 2012]) (SD complete). For the essentiality test in minimal media, the same SD was used, but only supplemented with histidine 3.5 mg/liter (#DOC0142; Formedium), methionine 4 mg/liter (#DOC0166; Formedium), and uracil 4 mg/liter (#DOC0212; Formedium), required for the survival of our strains given their genetic background (SD minimal). For the essentiality test in respiration media, we used YPGlycerol, prepared in the same manner as YPD but with 3% [wt/vol] glycerol (CAS56-81-5; J.T. Baker: Product code: 15557974) instead of glucose. In all cases, 5-Ph-IAA (Inducer for AID2 System; CAS: 168649-23-8; Bio Academia: Product code: 30-003-10) was used at a working concentration of 5 µM and added to the media for the time indicated in each figure. A stock solution (100 mM in DMSO/20,000X) was kept in aliquots at –20°C and only thawed/frozen for a few cycles. For solid media, the 5-Ph-IAA was incorporated together with the antibiotics when the temperature of the media was sufficient to keep the agar liquid, but low enough to prevent damaging the 5-Ph-IAA. Utilizing a working concentration of 1 µM provided the same results as 5 µM in time course analysis, data not shown.

**Yeast library generation**
The C′ AID-GFP SWAT library generation was performed as described ([Weill et al., 2018]). In short, a RoToR array pinning robot (Singer Instruments) was used to mate the donor strain (Table S4) with the parental C′ SWAT library. The mating was followed by sporulation and selection to generate the desired haploid α library, harboring all the required elements ([Cohen and Schuldiner, 2011]; [Tong and Boone, 2007]): NAT-TEF2p-Os-TIR1(F74G), XXX(ORF)-AID*-eGFP::G418. For SceI-mediated tag swapping, the library was grown on YPGalactose (2% [wt/vol] peptone, 1% [wt/vol] yeast extract, 2% [wt/vol] galactose) and then moved to SD containing 5-fluoroorotic acid (5-FOA, Formedium) at 1 g/liter. Finally, NAT and G418 were employed to select the strains that successfully completed the SWAT procedure. Images of the finished library were recorded with a CCD camera (Amersham ImageQuant 800 systems; Cytiva). Images were analyzed with SGA tools ([Wagih et al., 2013]) to extract the colony sizes and compared with the images of the parental library (C′ SWAT) to determine colony survival.

**Library validation by targeted sequencing (Anchor-seq)**
The Anchor-Seq method ([Meurer et al., 2018]) for targeted sequencing of the pooled library was utilized. In brief, yeast strains from a grown agar plate of the library were collected, washed with buffer, aliquoted (100–150 mg per tube), and stored at –20°C until use. The Zymo Research YeaStar kit (cat. No.: D2002) was used as per the manufacturer's directions to extract genomic DNA from one aliquot per plate, with a typical yield of 115–125 ng/µl in 60 µl. To fragment the DNA, the NEBNext UltraTM II FS DNA Library Prep Kit for Illumina (Cat. No.: 7805S) was used with fragmentation lasting for 5 min at 37°C. The bubble-shaped adaptor was achieved by resuspending oligos #1 and #2 (Table S5) in TE buffer and annealing them in annealing buffer via heat denaturation and slow cooling. The fragmented DNA received 2.5 µl of the bubble-shaped adaptor at a concentration of 15 mM that was ligated for 15 min at 20°C in a LAB cycler (a SensoQuest PCR model). NEBNext Sample Purification SPRI Beads were used for size selection and cleanup of adaptor-ligated DNA. A selective PCR preformed with oligos #3 and #4, was performed for 18 cycles, followed by a new round of beads purification, and a second PCR with primers #5 and #6, for 15 cycles. Final size selection and cleanup were performed before DNA quantification with a Qubit Flex Fluorometer (Cat. No. Q33327; Thermo Fischer Scientific) and normalization to 8 nM per tube. Equal volumes from the four normalized samples were pooled together and spiked with 25% Phi X DNA before being subjected to NGS sequencing. 15. NGS. We used Illumina MiSeq, v3_600 cycles flow cell kit (Cat. No. MS-102-3003), with a

paired-end protocol allocating 310 read cycles from the Rd1 and Rd2 directions, and six bases read for the Index 1 barcodes. The barcodes assigned to each sample (refer to Table S5) were utilized to match the reads with their respective 1,536 plate. Sequencing and Bioinformatic analyses were performed at the Mantoux Bioinformatics G-INCPM, the Weizmann Institute. FASTQ files accession number: PRJNA1121868.

## Automated high-throughput fluorescence microscopy

For the short-term microscopy screen, the library was transferred from agar plates into 384-well plates for growth in 100 µl YPD media with antibiotics overnight (ON) at 30°C using the RoToR arrayer. The ON culture was back-diluted into 384-well plates in YPD to an $OD_{600}$ of ~0.2. After 4 h, 50 µl from each well were transferred to a glass-bottom 384-well microscopy plate (Azenta Life Sciences) coated with concanavalin A (Sigma-Aldrich). The logarithmic phase cultures were allowed to bind to the bottom of the plate for 20 min. After incubation, wells were washed twice with SD complete media to remove non-adherent cells. The plates were then transferred to an Olympus automated inverted fluorescent microscope system using a robotic swap arm (Peak Robotics). Cells were imaged at room temperature (RT) in SD complete media using an Olympus UPLFLN 60X air lens (NA 0.9), a CSU-W1 Confocal Scanner Unit (Yokogawa), equipped with a 50 µm pinhole disk, and with an ORCA-flash 4.0 digital camera (Hamamatsu), using the ScanR software (version 3.2). Images were recorded with 488 nm laser illumination for the GFP channel (excitation 488 nm, emission filter 525/50 nm) with 500 ms exposure and 70% laser intensity, mild conditions to minimize photobleaching (compared with long-term screen below). Brightfield images were also taken, with 200 ms exposure and 100% halogen bulb. Two positions were imaged per well and the software autofocus was used to ensure the cells were imaged in their central plane for proper comparison. After imaging, the plates were removed from the microscope, 5-Ph-IAA was added to a final concentration of 5 µM per well, and the plate was incubated for 30 min at RT. After incubation, the plate was re-imaged as previously. The same positions were revisited. All the steps in the automated imaging process were done by an automated imaging platform EVO freedom liquid handler (TECAN).

For the long-term microscopy screen, the procedure was as described above, with the changes here noted. The back dilution culture (100 µl) was split into two: 50 µl was transferred to the microscopy plate and imaged (before induction), while the remaining 50 µl received 5-Ph-IAA to a final concentration of 5 µM. These plates were grown for 20 h at 30°C to become the ON plate for the induced cultures. After 20 h, 5 µl were back-diluted into 95 µl of SD complete with 5 µM 5-Ph-IAA. After 4 h of back dilution, 50 µl from each well was transferred to the microscopy plate. Cells were allowed to bind to the bottom of the plate for 15 min. After incubation, the media was replaced by fresh media, with 50 nM MitotrackerOrange CMTMRos (Thermo Fisher Scientific) and 5 µM 5-Ph-IAA. The dye was incubated for 15 min and washed with SD complete with 5 µM 5-Ph-IAA. Cells were imaged as described previously in SD complete with 5 µM 5-Ph-IAA (24 h induced cells). Since the cells were not

re-imaged, harsher illumination conditions were used to improve GFP visualization (1,000 ms exposure time, 80% laser intensity). In the induced case, MitotrackerOrange was imaged with an exposure of 600 ms, 60% laser intensity (excitation 561 nm, emission filter 617/73 nm).

## Low-throughput fluorescence microscopy

Images displayed in Figs. 2 and 4 and Figs. S3 and S4 were acquired in the same setup as before but with an Olympus UPLFLN 100× oil objective (NA 1.3).

Images displayed in Fig. 1 were acquired at RT using a Visi-Scope Confocal Cell Explorer system, composed of a Yokogawa spinning disk scanning unit (CSU-W1) coupled to an inverted Olympus IX83 microscope with a 60× oil objective (NA 1.4). The excitation wavelength was 488 nm, with an exposure time of 600 ms and 80% laser intensity). Images were taken by a PCO-Edge sCMOS camera controlled by VisiView software (Visitron Systems version 3.2.0).

The microscopy images were cropped and colored in ImageJ (Schneider et al., 2012). The brightness and contrast of images were linearly adjusted so that all GFP images of the same strain (before and after induction) have the same parameters. For MitoTracker images, all the images in the same panel, irrespective of the strain or induction state, share the same parameters. In cases where the cell boundaries were not visible, the Dotted line plugin for ImageJ was used to denote the cell edge.

## Essentiality assays

The C′ AID-GFP library was replicated in a 1,536 format in agar with antibiotics, using the RoToR arrayer (Singer). Two copies were replicated in each media tested (YPD, SD minimal and YPGlycerol, see yeast growth) with 5 µM 5-Ph-IAA and two replicas without. After 24 h of incubation at 30°C (or 48 h for YPGlycerol), cells were refreshed into new plates using the RoToR arrayer. After a second period of incubation (24 h at 30°C, 48 h for YPGlycerol), all the plates were imaged with a CCD camera (Amersham ImageQuant 800 systems; Cytiva). Images were analyzed with SGA tools (Wagih et al., 2013) to extract the colony sizes. An average of both replicates was used to calculate the ratio between the colonies grown with 5-Ph-IAA and the colonies grown without (not induced). As controls, a WT strain (YMS3553, Table S2) and three library strains known to give induction-provoked lethality were incorporated into each plate in triplicates. Since our metric was based on a ratio between induced and uninduced strains, positional effects (such as better growth in the plate edges) are expected to cancel out. However, we normalized the scores of each plate and each media to facilitate comparisons using the median value per plate, after removal of outliers.

## Growth assays

Cells were grown ON in YPD with antibiotics and the culture was washed from old media and back diluted to $OD_{600}$ 0.2 in fresh YPD containing or not 5 µM 5-Ph-IAA. Cells were incubated at 30°C with agitation and growth was monitored every hour for 24 h with a Spark multimode microplate reader

(TECAN) by measuring the $OD_{600}$. In the cases of the GAL promotor swap, the ON growth in YPD served to deplete the mRNA from the cells. The cells were then grown on YPD (repressive conditions) or YPGalactose (prepared as YPD but with 3% galactose). At least three replicates were used in every case.

### Responsiveness under different conditions

Protein depletion was successfully verified under different media conditions after 45 min of induction, including SD (MSG) complete (prepared as SD complete but 0.1% MSG (Minerva); 0.17% YNB without amino acids and without ammonium sulfate (Minerva); SD (MSG) complete with 5 µg/ml Terbinafine (Minerva) or 2 mM DTT (Sigma-Aldrich); SD (MSG) complete without thiamine (Minerva); without biotin (Minerva); without copper (Minerva); without iron (Minerva) or without phosphate (Minerva); SGalactose (MSG) complete; DDW and conditioned media SD (MSG) complete after 48 h of yeast growth, followed by sterilization by addition of antibiotics.

### Creation of grid images of individual yeast cells

To maximize the information content of each microscopy image, single yeast cells were automatically segmented from the brightfield image using a neural network within ScanR Analysis software. Detected cells were individually cropped by delimiting a square bounding box of 65 × 65 pixels centered on them. A montage image assembled up to 400 individual cells from the same strain laying them out in a grid of 20 by 20 cells. Thus, for each strain, we generated a cell grid image of dimension 1,300 × 1,300 pixels for each combination of conditions tested (short versus long induction and before versus after induction). In addition, cell masks were transferred from brightfield ("TransCon") across the fluorescent channels (488 and 561 nm) to yield one cell grid montage per channel. To facilitate visualization, the contrast of each cell-grid image was adjusted such that 0.35% of pixels were saturated. Importantly, we preserved fluorescence intensities across cells, thus allowing us to assess for noisy protein expression (e.g., high variance across cells) or intensity variations between timepoints (e.g., before or after 30 mn/24 h induction). To save storage space and increase performance in loading images, intensities of brightfield images were scaled down to 8 bits since intensity variations were not useful. Custom Python scripts to generate cell grids are available in https://github.com/benjamin-elusers/ydimmer.

### Analysis displayed in images

#### "SWATting" survival

Images of the C′ SWAT before library creation and of C′ AID-GFP in agar plates were quantified with SGA tools (Wagih et al., 2013), to estimate the colony "SWATting" survival. A colony was considered present if the quantification was over 50 A.U. Fig. S1 A.

#### "SWATting" efficiency

The top 500 most abundant proteins (Ho et al., 2018), whose strains were present in the library, were evaluated for fluorescence. Since these highly abundant proteins should provide a clear signal, the percentage of strains with a signal detected can serve as a proxy for "SWATting" efficiency, meaning the genetic modification with the AID tag and GFP module was successful. Fig. S1 B.

### Sequencing validation and coverage

From the sequencing output (see Library validation by targeted sequencing [Anchor-Seq]), we classified the strains into categorical results. The categories represent whether the strain was validated by sequencing, not found, or found with an undesired modification (mutation). The percentage of correct sequences over the total found gave us the sequencing validation, Fig. S1 C; while the percentage recovered over the submitted gave us the coverage, Fig. S1 D.

### Fluorescence and responsiveness classification

Images acquired in the long and short induction screen were processed with ScanR Analysis to segment individual cells and measure the mean green fluorescence per cell. For each strain, cells with signals outside of a range of two standard deviations (SD) were removed and the geometric mean, SD, and number of cells ($n$) were reported, Table S1. Strains where the fluorescence is bigger than the brightest non-fluorescent control were analyzed in a quantitative manner to determine responsiveness (significant GFP signal reduction after induction). For the remaining strains, mainly organellar ones, visual examination was used to determine if the GFP signal was present and if it was reduced upon induction. Values and individual results for each strain are listed in Table S1, while the aggregate of the annotations was utilized to calculate the percentages displayed in Fig. 2 A. The individual values for fluorescent strains were plotted in Fig. 2 B (long induction) and Fig. S2 B (short induction) in RStudio (Posit team, https://posit.co/, 2023) with the ggplot2 package (Wickham, 2016).

### Responsiveness across cellular compartments

The subcellular protein localization for each fluorescent strain in the library was adapted from Huh et al. (2003) to include only the first category listed and only the categories listed in Fig. 2 D. For that, "early Golgi" and "late Golgi" were considered together with "Golgi"; "nucleolus" was integrated to "nucleus"; and "actin," "endosome," "ER to Golgi," "microtubule," and "spindle pole" were considered under "punctate composite." The percentage of responsive strains for each category is displayed in Fig. 2 D.

For mitochondrial subcompartments, the annotations from Williams et al. (2014) were taken and used to compute the percentage responsiveness across mitochondrial compartments, Fig. 2 E, and further considering co- versus posttranslational translocation, Fig. S2 C.

For bias in responsiveness according to orientation, the localization of the C′ of the ER proteins (as determined by Huh et al. [2003]) was computed by the majority rule from the algorithms listed in Topology yeast (Weill et al., 2019). Annotations are listed in Table S1, and the group percentage of responsiveness is displayed in Fig. S2 D.

### Fluorescence and responsiveness across protein abundance

All strains present in the library were segregated by protein abundance (Ho et al., 2018) into deciles, with decile 1 clustering the most abundant proteins and decile 10 the least. The percentage of the strains of each decile with detected fluorescence or responsiveness are presented in Fig. S2, E and F, respectively.

### Go slim analysis for proteins essential under different conditions

The list of strains with relative growth below 0.5 in induced versus uninduced conditions (hits) in each media were taken to generate the Venn diagram displayed in Fig. 2 C. All the GO slim terms (Ashburner et al., 2000; Gene Ontology Consortium et al., 2023) associated with each of the hits were grouped (i.e., minimal media includes 480 hits). Since most hits were shared (347 strains were this in every condition tested), to select the top three GO slim terms associated with each media, the exclusive hits were taken (meaning the strains that were only found to be essential in that specific condition, for minimal media, 73 out of 480 hits). For reporting the fraction of each GO slim term that was recovered, all the strains associated with that specific GO term were now utilized (i.e., "amino acid metabolic process," GO:0006520, was the term strongly associated with the hits exclusively in minimal media [22/73]; yet, if we consider all the 480 hits in minimal media, 27 hits are present). The percentage of the whole GO slim term annotations affected is reported, regardless of whether those strains were present or not in the library (continuing the example, "Amino acid metabolic process" has 151 genes associated with it, but 137 of those are represented in the C′ AID-GFP library, and 77 have been certified as responsive from fluorescent analysis; this gives a 17.9% of the total annotations for the term [27/151] as hits). The top three GO terms for each category are listed under the titles for each media in Fig. 2 C, with the percentage of each GO slim term recovered as hits on the screen. For the hits shared in all the conditions, only the 347 strains were considering when quantifying percentage of GO term recovery.

### Comparison with other systems

To compare the list of strains classified as essential in the YKO (Giaever et al., 2002; Winzeler et al., 1999) with the C′ AID-GFP results, the relative growth in rich media in induction versus not induction was considered. Venn diagrams displaying the overlap between the 560 essential genes defined in the YKO and defined as responsive in the C′ AID-GFP library and the strains with a relative growth below 0.5 (permissive threshold) or 0.1 (stringent threshold) are displayed in Fig. S3 A.

To compare the C′ AID-GFP library with other inducible systems, the list of proteins classified as essential in the YKO (Giaever et al., 2002; Winzeler et al., 1999) and present in the C′ AID-GFP library (irrespective of whether the strain is responsive), 650 proteins, was separated into quintiles, each with 130 proteins, by their mRNA abundance (Lipson et al., 2009), with quintile 1 representing the most abundant ones. A qualitative classification of the strains from the TET-off (Mnaimneh et al., 2004) and the YETI-E (Arita et al., 2021) collection into "responsive," "non-responsive," and "absent" allowed for the generation

of the scatter plots in Fig. S3 B of the percentage of genes with affected phenotypes versus mRNA abundance.

### Mitochondrial signal quantification and determination of MDM-like phenotypes

The images acquired of the MitoTracker dye after 24 h of induction (see Automated high-throughput fluorescence microscopy, long term) were processed as in *Fluorescence and responsiveness classification*, with a further segmentation of the mitochondria per cell done with the edge segmentation tool in ScanR Analysis. Changes in fluorescence intensity, mitochondria number, and relative growth in respiration media were instrumental in determining a strain as displaying MDM-like phenotypes, together with a blind manual examination of the images obtained, Table S1. The MitoTracker signal of each strain was ranked to generate the plot in Fig. 4 B.

### Ergosterol biosynthesis pathway proteins

Strains harboring the degron tag in a protein related to the ergosterol biosynthesis pathway were induced or not for 4 h and then imaged as described in low-throughput microscopy. The quantitation of GFP and MitoTracker fluorescence was done as detailed in fluorescence and responsiveness classification and displayed together with a schematic of the pathway adapted from Jordá and Puig (2020) in Fig. S4 C.

### Oxygen consumption rate

One day prior to measurement, a Seahorse XFe96/XF Pro Cell Culture microplate (Agilent) was coated with 0.1 mg/ml Poly-D-Lysine (Sigma-Aldrich) and incubated at 4°C overnight. Seahorse XF Calibrant (Agilent) was added to the Seahorse XFe96/XF Pro Sensor Cartridge plate (Agilent) and incubated overnight in a non-$CO_2$ incubator. Strains were grown ON in YPD with NAT and G418 at 30°C. On the day of the experiment, a back dilution to $OD_{600}$ of 0.5 was prepared in either YPD (not induced) or YPD containing 5 μM 5-Ph-IAA (induced) and incubated for 7 h at 30°C. The yeast cells were pelleted down at 500 $g$ after which they were resuspended in assay medium (0.67% YNB, 2% potassium acetate, and 2% ethanol) as described in Zhang et al. (2022), to an $OD_{600}$ of 0.1. 180 μl of cell suspension per well was added to the Seahorse XFe96/XF Pro Cell Culture microplate followed by incubation at 30°C for 30 min. Measurements were taken under basal conditions (from 0 min) and upon the addition of 20 μM CCCP (at 18 min) and 2.5 μM Antimycin A (at 36 min) in a Seahorse XFe96 Analyzer (Agilent). Three cycles of mixing (3 min) and measuring (3 min) time were allotted to each condition. Raw measurements are listed in Table S6. Respiration-related oxygen consumption was calculated by taking the basal oxygen consumption rate and subtracting the non-respiration oxygen consumption rate (measured with Antimycin A). Data analysis was done using Graph Pad Prism 10.2.3 and the data presented are the average of nine independent experiments. Graphs are plotted showing the standard error of the mean (SEM). Statistical significance was determined using two-way ANOVA with comparisons done using the Fisher Least Significant Different approach setting a false discovery rate to 0.05 with a two-stage linear step-up procedure of Benjamini,

Krieger, and Yekutieli. Significant results between the untreated control and any other group or between the untreated and treated group of the same strain are marked with a connecting line.

## Synchronized and altered morphology

Brightfield images acquired after 24 h of induction (see Automated high-throughput fluorescence microscopy, long term) were manually examined to determine candidates displaying synchronized and/or altered morphology, Table S1.

## Western blot

Cells were grown ON in YPD with selections and used to inoculate 125 ml of YPD to $OD_{600}$ 0.05. Cells were incubated with shaking at 30°C until $OD_{600}$ 0.2, where they were spit into 18 identical cultures. The first series of technical replicates received 5 µM 5-Ph-IAA (time point 180 min) and the subsequent groups received it after 1.5, 2, 2.5, and 2.8 h so that after 3 h of growth all the cells were ready for harvesting. Cells were harvested by centrifugation, followed by flash-freeze in liquid nitrogen. Protein extraction, SDS-PAGE, and western blotting were performed as described previously (Eisenberg-Bord et al., 2021). Briefly, the cells were lysed with 8 M urea-based lysis buffer with protease inhibitors and glass beads-beating (Scientific Industries). Lysates were denatured by the addition of SDS (final concentration ∼2%) and a 45°C-incubation for 15 min. Denatured lysates were centrifuged to separate cell debris. Loading buffer containing DTT (final concentration ∼25 mM) and incubated at 45°C for 15 min. 30 µg sample was loaded onto 12% agarose gels and separated with electrophoresis, then transferred onto nitrocellulose membrane using the Trans-Blot Turbo transfer system (Bio-Rad). Membranes were blocked in SEA BLOCK buffer (Thermo Fisher Scientific), incubated with primary antibodies (anti-GFP, 1:1,000, ab290; Abcam and Anti-actin, 1:5,000, ab170325; Abcam), washed, and incubated with fluorescent secondary antibodies for 1 h (926-32210, 1:10,000; Li-COR and ab216777, 1:10,000; Abcam). After washing, the membranes were imaged on the LI-COR Odyssey Infrared Scanner. Images were analyzed with GelAnalyzer 23.1.1. Automatic lane detection was used, followed by the rolling ball method to define the baseline, and automatic peak detection. Raw volume measurements were used to quantify the protein band and the loading control (Act1). Each peak was normalized by the corresponding Act1 band. For comparison with the microscopy method, each replicate was normalized to its corresponding starting time (100% signal).

## Flow cytometry

The strains were grown ON at 30°C and shaking, 180 RPM (New Brunswick model Innova 44) in Falcon 15-ml rounded-bottom tubes (Corning) with a loose cap with 1 ml of S.D. complete [MDRV1] medium supplemented with Nat antibiotics. The O.N. cultures were back-diluted into 2 ml of the same media to an $OD_{600}$ of 0.2. The cultures were allowed to grow until reaching $OD_{600}$ of 0.5 when they split into half. In one half of each sample, 5 µM 5-Ph-IAA was added before incubation was continued for another 3 h and then stopped by adding 2 ml of ethanol (final

concentration 66%). After ethanol addition, the cultures' tubes were vortex-mixed rigorously before being stored at –20°C until further use (a minimum of ON). The cell cultures were prepared for DNA visualization by flow cytometry analysis as previously described (Rosebrock, 2017). In short, following ethanol fixation, cells were treated with RNAse A (Sigma-Aldrich) to remove RNA and, after that, with Proteinase K (Sigma-Aldrich) to remove proteins. The treated cell samples were then stained with 2.5 µM Sytox Green (Thermo Fischer Scientific) to visualize DNA content before analyzing the samples at the Flow Cytometry Core Facility of the Weizmann Institute of Science [MDRV2] [MDRV3] using the BD FACSAria III Cell Sorter equipped with a 100-nm nozzle and controlled by BD FACS Diva software v8.0.1 (B.D. Biosciences). DNA content of the cell population was collected by excitation of the Sytox Green at 488 nm and monitoring its emission at 502 with long-pass and 530/30 bandpass filters and plotted as the area of the Sytox-green signal versus cell count. Further analyses were performed using FlowJo software v10.2 (Tree Star).

## Online supplemental material

Fig. S1 shows the performance metrics from the creation of the AID library. Fig. S2 shows the AID library performance according to different parameters. Fig. S3 shows the comparison of the AID library and other systems. Fig. S4 shows the effect of depleting proteins with the AID system on known MDM strains and the ergosterol pathway. Fig. S5 shows Cdc48 and Npl4 upon protein depletion. Fig. S6 shows images from the yDIMMER website, which allows visualization of the strains present in the AID, after short and long depletion times. Table S1 lists all the strains present in the AID library and their features as used in the different analyses of this work. Table S2 contains all yeast strains used in this study. Table S3 contains all plasmids used in this study. Table S4 contains all primers used in this study, except the primers used for the library sequencing. Table S5 contains the oligos and primers used for the Anchor-seq protocol (library sequencing). Table S6 contains the oxygen consumption rates for the strains tested (Figs. 4 and S4).

## Data availability

Sequencing data can be found with the FASTQ files accession number: PRJNA1121868. Processed and raw images are available at https://www.weizmann.ac.il/ydimmer/. Metadata can be found in the supplementary material. Further material can be provided upon reasonable request.

## Acknowledgments

We thank Noga Preminger and Naama Zung for the critical reading of the manuscript. We thank Nir Friedman and Daphna Joseph-Strauss for their help with plasmids. We thank Michael Knop for generously sharing the C′ SWAT library and donor plasmids. We thank Emmanuel Levy for the reagents and Naama Zung for the Upc2-GFP strain. We are thankful for the amazing work of our support team: Hadar Meyer, Hanni Naor, and Reut Ester Avraham; the sequencing unit team and Amir Szitenberg from the Mantoux Bioinformatics G-INCPM unit, at

the Weizmann Institute for their service with the library sequencing; Ronen Hayun, Eli Hotoveli, Arie Pinto, and Maayan Maron from the Internet and Mobile Section for their service developing the yDIMMER website and Assaf Glik and Lior Michael from the server unit for providing safe storage and access to the yeast cell images.

This study was supported by the Minerva foundation, a Chan Zuckerberg Initiative (CZI) grant (2023-331952), and by the Institute for Environmental Sustainability (IES) at the Weizmann Institute of Science. The work on protein targeting in the Schuldiner lab is supported by a European Union ERC CoG (On-Target 864068). Collaborative research of the M. Schuldiner as P. Rehling labs is supported by the Deutsche Forschungsgemeinschaft SFB1190 (P11, P13). Supported by the Max Planck Society (P. Rehling) and the DFG under Germany's Excellence Strategy - EXC 2067/1- 390729940. This research was also supported by the Institute for Environmental Sustainability (IES) and the Knell Family Center for Microbiology at the Weizmann Institute of Science. R. Valenti was supported by the Alvin, Myra, and David Kaye memorial award for excellence in Life Sciences related research for international students. The robotic system in the Schuldiner lab was purchased through the kind support of the Blythe Brenden-Mann Foundation. M. Schuldiner is Incumbent of the Dr. Gilbert Omenn and Martha Darling Professorial Chair in Molecular Genetics.

Author contributions: R. Valenti: Data curation, Formal analysis, Investigation, Methodology, Supervision, Visualization, Writing - original draft, Writing - review & editing, Y. David: Investigation, Methodology, Writing - review & editing, D. Edilbi: Formal analysis, Investigation, Validation, Writing - review & editing, B. Dubreuil: Data curation, Formal analysis, Software, Writing - review & editing, A. Boshnakovska: Formal analysis, Investigation, Writing - review & editing, Y. Asraf: Data curation, Investigation, Methodology, Software, Writing - review & editing, T.-M. Salame: Formal analysis, Investigation, Resources, E. Sass: Investigation, Methodology, P. Rehling: Data curation, Funding acquisition, Project administration, Resources, Supervision, Writing - review & editing, M. Schuldiner: Conceptualization, Funding acquisition, Project administration, Resources, Supervision, Writing - original draft.

Disclosures: The authors declare no competing interests exist.

Submitted: 19 September 2024

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

# Supplemental material

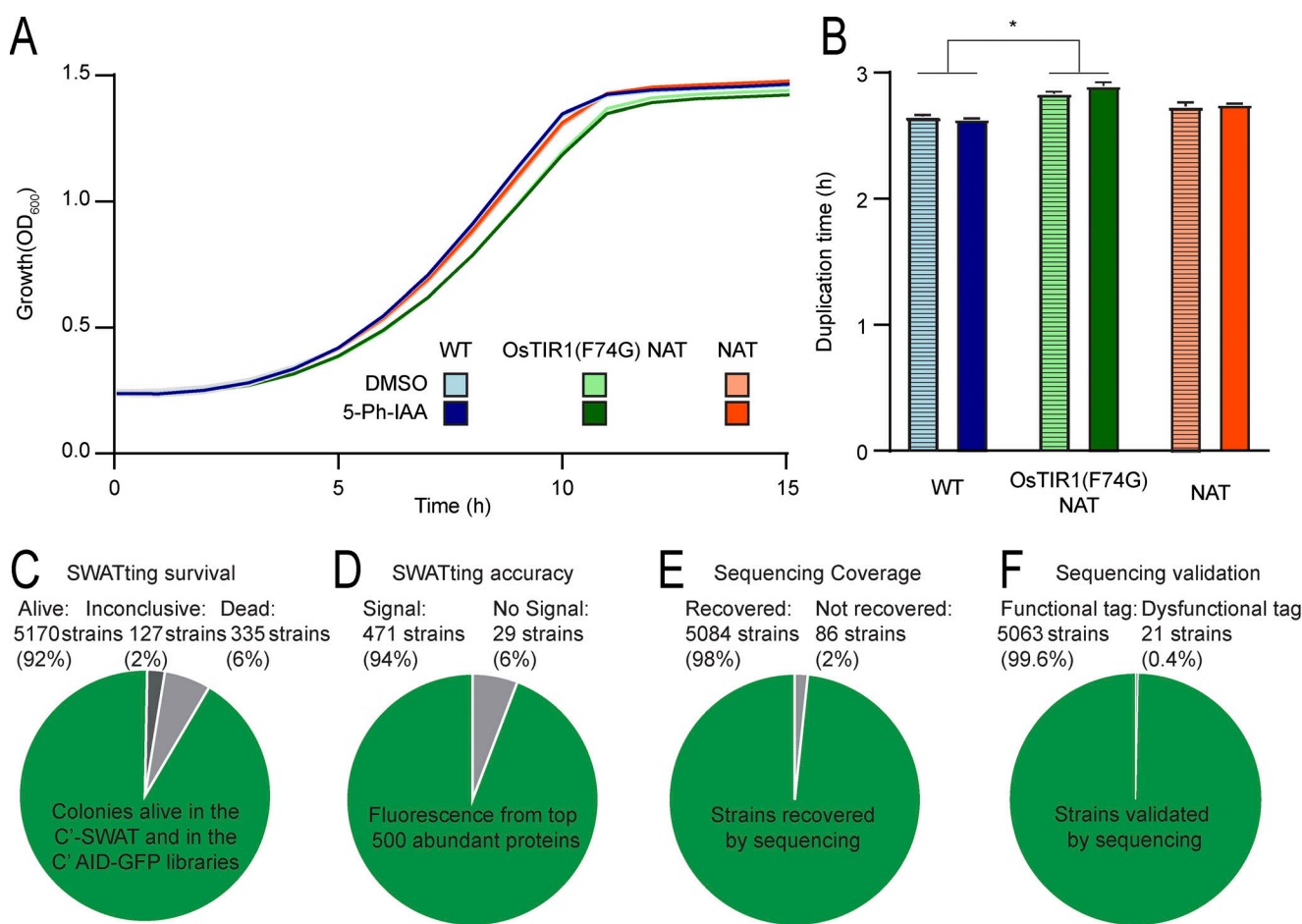

Figure S1. **The C' AID-GFP library extends the degron approach to a proteome level. (A)** Growth curves of control (WT) strain (blue, BY4741), the same strain transformed with the OsTIR1(F74G) and resistance to nourseothricin (NAT, green), or WT transformed only with the resistance to NAT (orange); all with 5 µM 5-Ph-IAA in DMSO (dark colors) or DMSO only (light colors). The strain containing the OsTIR1(F74G) presents a small growth defect when compared to WT or to the strain with matching resistance. **(B)** Bar graph displaying the duplication time calculated from the result from panel A. From a two-way ANOVA, the strain expressing the OsTIR1(F74G) grew at a significantly slower rate than the WT strain (P value 0.03, difference 8.3 ± 0.1%). Treatment with 5-Ph-IAA did not produce any effect. **(C)** Pie chart displaying the rate of survival of strains following the SWATting procedure. Colony presence was measured for the parental (C' SWAT library) and derived (C' AID-GFP) libraries, to calculate the 92% survival rate. **(D)** Pie chart displaying the fluorescent status of the top 500 most expressed proteins in the cell (Nash et al., 2020). For such abundant proteins a fluorescent signal was expected, therefore the estimation of 94% SWATting efficiency corresponds to the strains displaying a fluorescence signal. **(E)** Pie chart displaying the sequencing coverage (sequences from the whole library recovered by Anchor-Seq method), which was 98%. **(F)** Pie chart displaying the proportion of functional tags detected. Of the 5,084 strains detected by pooled sequencing, only 21 included variants that could compromise the function of the AID tag.

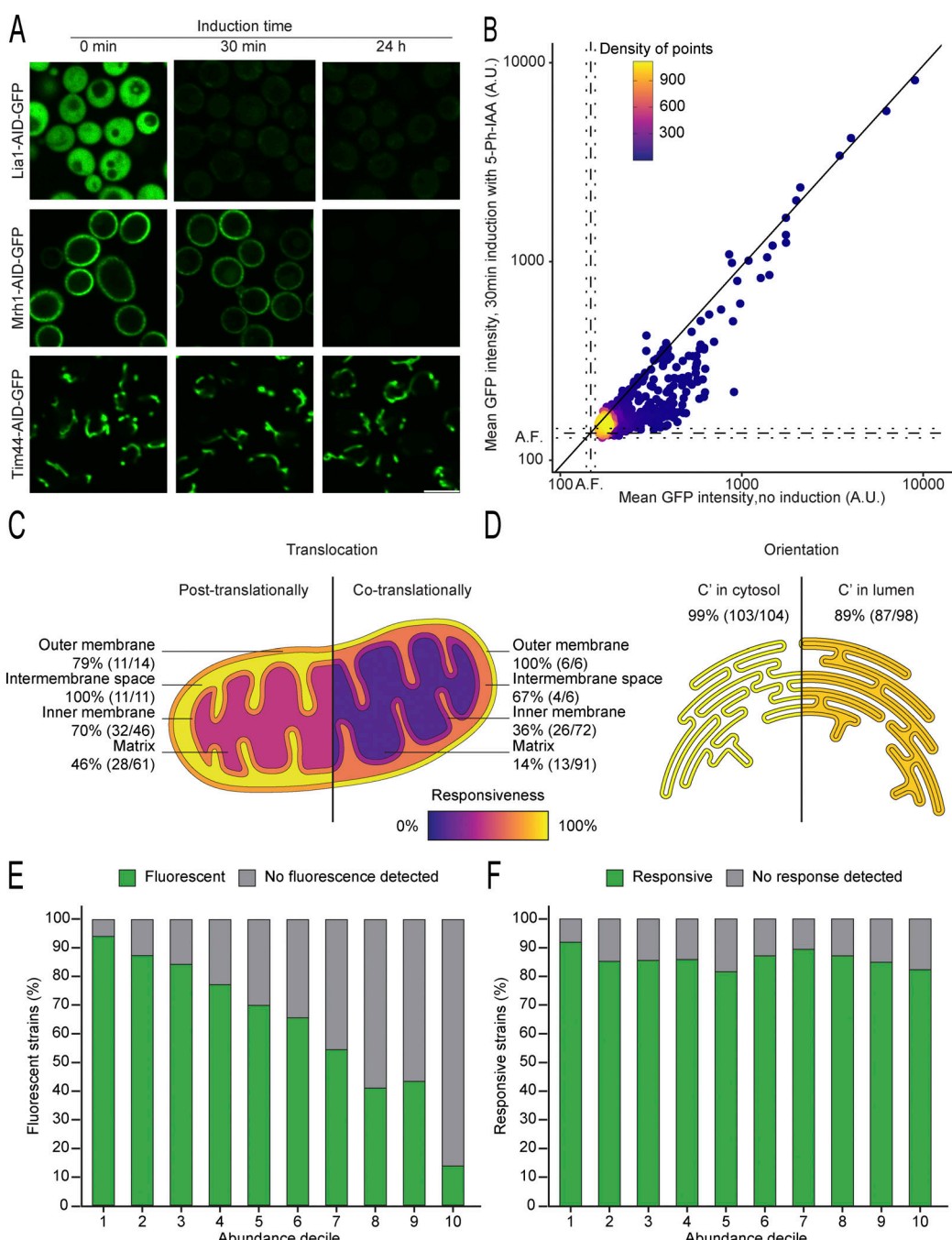

**Figure S2. Characterization of responsiveness across different parameters highlights the factors influencing the C' AID-GFP efficiency. (A)** Fluorescent images of selected proteins were imaged after no (0 min), short (30 min), or long (24 h) periods of induction. Different strains respond to the treatment with different kinetics, with some being depleted below detection after short induction times (Lia1), while others require longer induction times but were below detection levels after 24 h of induction (Mrh1). In cases where the signal is too dim, the outline of the cells is depicted with a dotted line. Scale bar: 5 μm. **(B)** Dot plot showing the mean GFP fluorescence intensity per cell for each strain before or after a short induction time. The background autofluorescence is marked in dotted lines (A.F., mean and two standard deviations in each direction). The diagonal line indicates the expected location for irresponsive strains. Already in this short induction time, there is a clear response in many strains. **(C and D)** Schematics depicting the differences in responsiveness for proteins that localize to mitochondria (panel C) and are co- versus posttranslationally translocated (Williams et al., 2014), or to the endoplasmic reticulum (panel D) and have their C' exposed in the cytosol or hidden in the lumen. Proteins targeted to mitochondria only after their synthesis is complete (post translational) are more likely to be depleted below detection levels upon induction of the AID system. Proteins with the C' exposed to the cytosol are also more likely to be degraded than their counterparts with hidden C'. This reflects differences in the accessibility of the AID tag to the TIR1 adaptor protein, the ubiquitination machinery and the proteasome, required for its induced depletion. Translocation and topology information were taken from the literature (Jan et al., 2014; Weill et al., 2019). **(E)** Bar plot of percentage of fluorescent strains versus abundance decile, with decile 1 being the most abundant proteins (Nash et al., 2020). As expected, abundant proteins are more likely to have a fluorescent signal. **(F)** Bar plot of percentage of responsive strains versus abundance decile (Nash et al., 2020). For the fluorescent strains in each decile, no clear correlation between abundance and responsiveness is detected. Responsiveness remains above 80% for all the deciles.

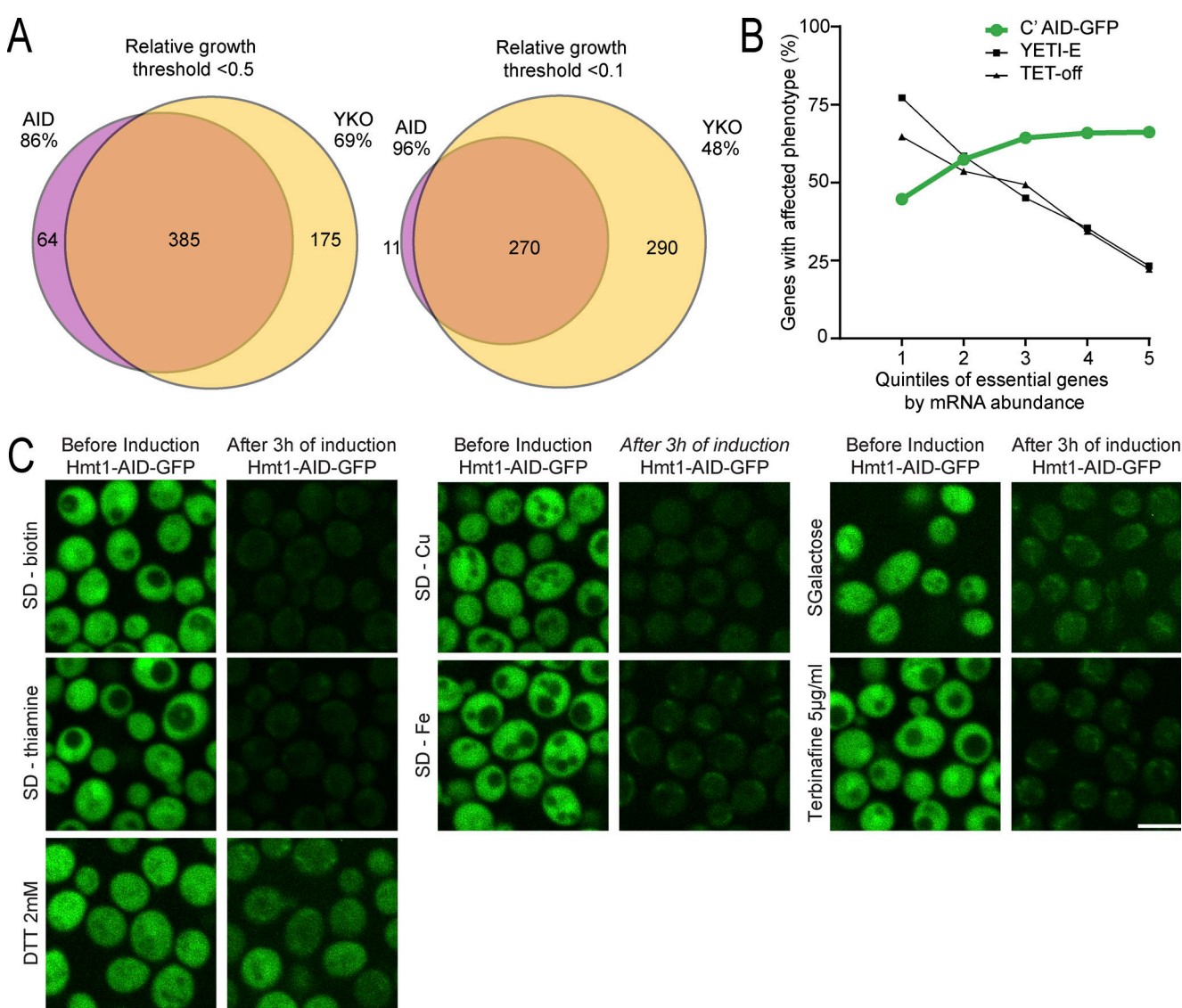

Figure S3.    **The C' AID-GFP library is complementary to other approaches. (A)** Venn diagrams comparing the results from the C' AID-GFP library with the YKO collection (Giaever et al., 2002) for hits defined with permissive (<0.5 relative growth) or stringent (<0.1 relative growth) thresholds. **(B)** Graphs showing the comparison of the C' AID-GFP library versus other inducible collections such as the YETI-E (Arita et al., 2021) and TET-off collection (Mnaimneh et al., 2004) across protein abundance, where quintile 1 to 5 group the proteins by mRNA levels from the most to the least abundant (Lipson et al., 2009). **(C)** Fluorescence microscopy images demonstrate that the AID system works robustly under different conditions. Scale bar: 5 µm.

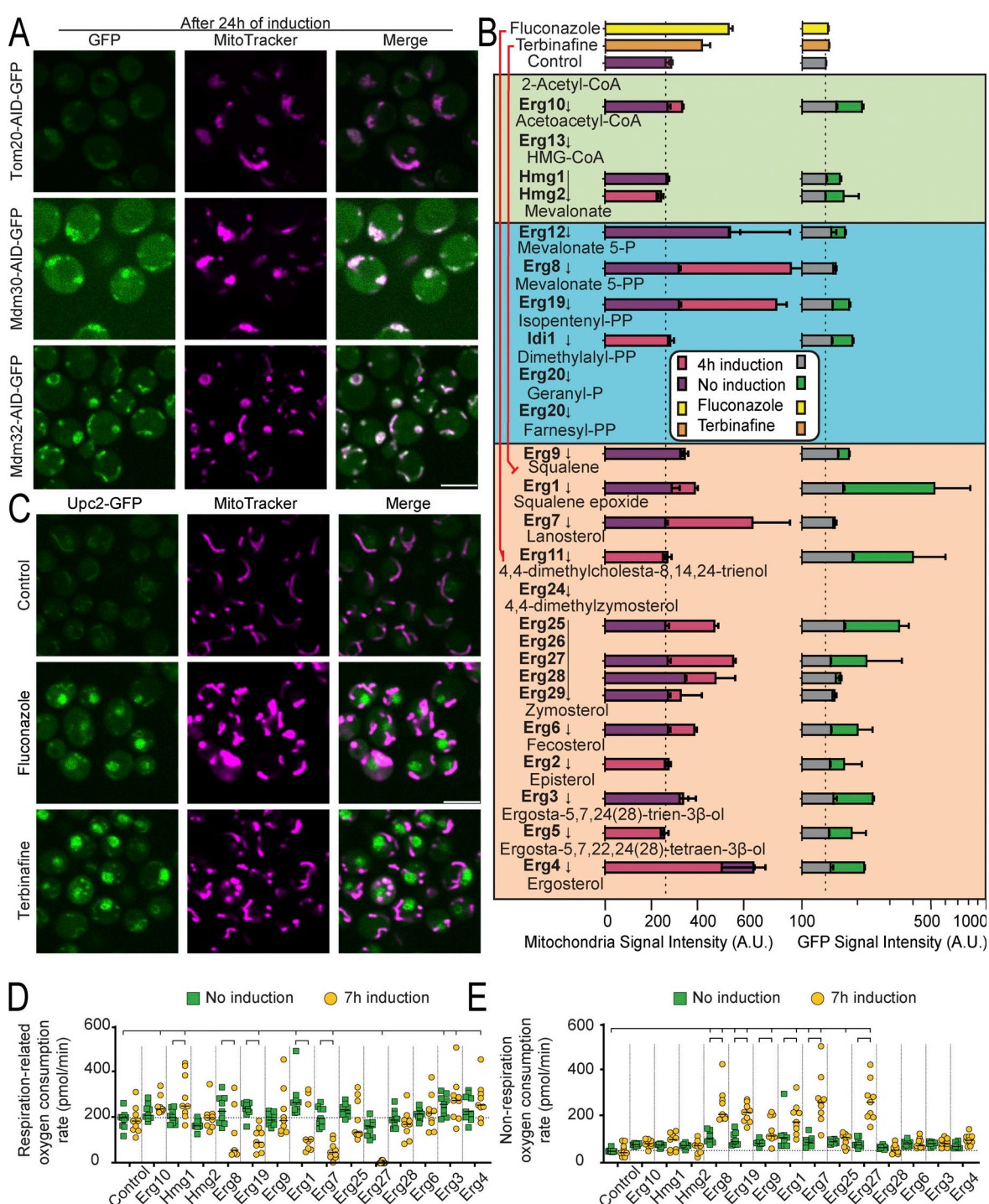

Figure S4. **Known proteins and pathways required for proper mitochondrial morphology and distribution display altered mitochondrial phenotypes as expected. (A)** Fluorescent images of known MDM strains were imaged after 24 h of induction. The mitochondria morphology (MitoTracker channel) is severely affected after protein depletion (GFP channel). Scale bar: 5 µm. **(B)** Schematics of the ergosterol biosynthesis pathway with quantification of the AID strains for the MitoTracker signal (purple no induction, magenta 4 h induction) and GFP (green no induction, gray 4 h of induction). Dashed vertical lines indicate the background fluorescence in the control. Changes are observed irrespective of whether the proteins affected participate in the pre (blue and green) or post (orange) squalene module of the pathway. Image adapted from Jordá and Puig (2020). **(C)** Fluorescent images of a strain containing the ergosterol sensitive protein, Upc2, C′ tagged with GFP under control conditions, or upon 4 h of treatment with fluconazole or terbinafine, which inhibit ergosterol biosynthesis. Upc2 translocates to the nucleus (green) upon ergosterol depletion. Concomitantly, the MitoTracker signal intensity increases and the mitochondrial morphology changes. **(D)** Bar graph showing the oxygen consumption rate derived from respiration for a control and selected strains, before and after 7 h of induction with 5-Ph-IAA. Significant changes for each group with the control and for each induced versus their corresponding uninduced group are marked (one-way ANOVA, FDR 0.05). **(E)** Bar graph showing the oxygen consumption rate not derived from respiration, before and after 7 h of induction with 5-Ph-IAA. Several strains present increased oxygen consumption. Significant changes for each group with the control and for each induced versus their corresponding uninduced group are marked (one-way ANOVA, FDR 0.05).

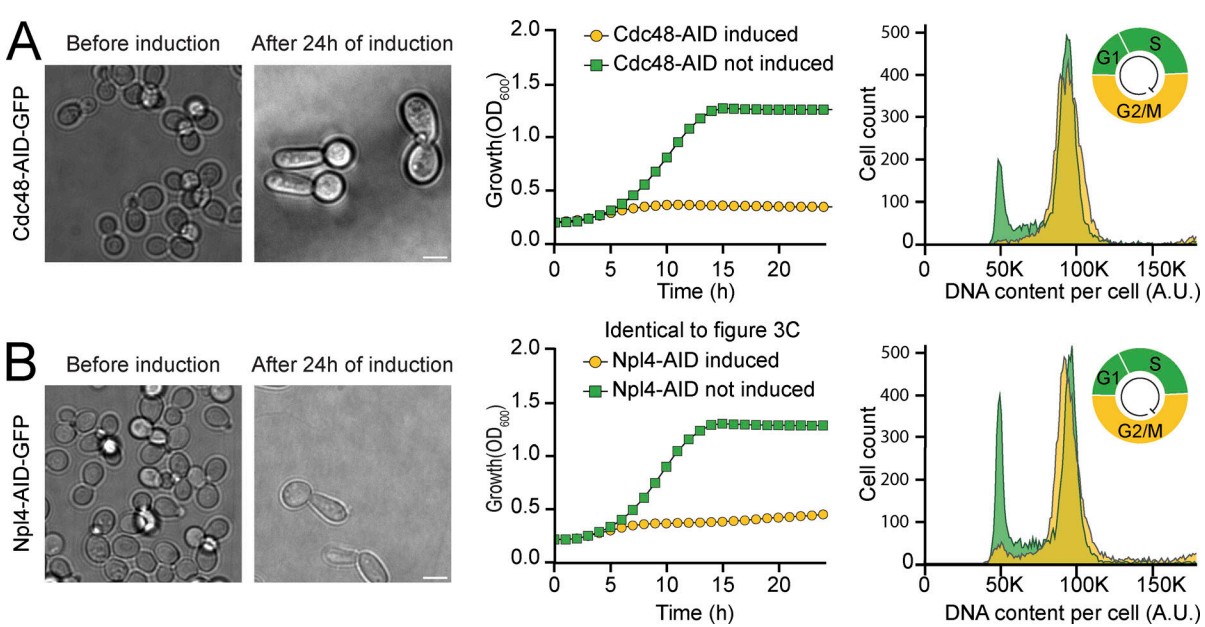

Figure S5. **A Cdc48 complex member displays CDC characteristics. (A)** Left: Brightfield microscopy images of Cdc48, before and after induction of the degron system for 24 h. A distinct and synchronized altered morphology can be observed after protein depletion as previously shown. Scale bar: 5 µm. Center: Growth curves of Cdc48, with induction of the degron (gray curve) or not (green curve). Lethality is seen upon depletion of Cdc48. An average of three repeats with their standard deviation is shown. Right: Flow cytometry measurements for Cdc48, with or without 4 h induction. Without induction (green), a typical histogram for cycling cells is appreciated, while an arrest in G2/M stage can be seen upon protein depletion (gray and upper right corner schematic). **(B)** Same as panel A, but for Npl4, a Cdc48 interactor. The growth curve is identical to the one displayed in Fig. 3 C.

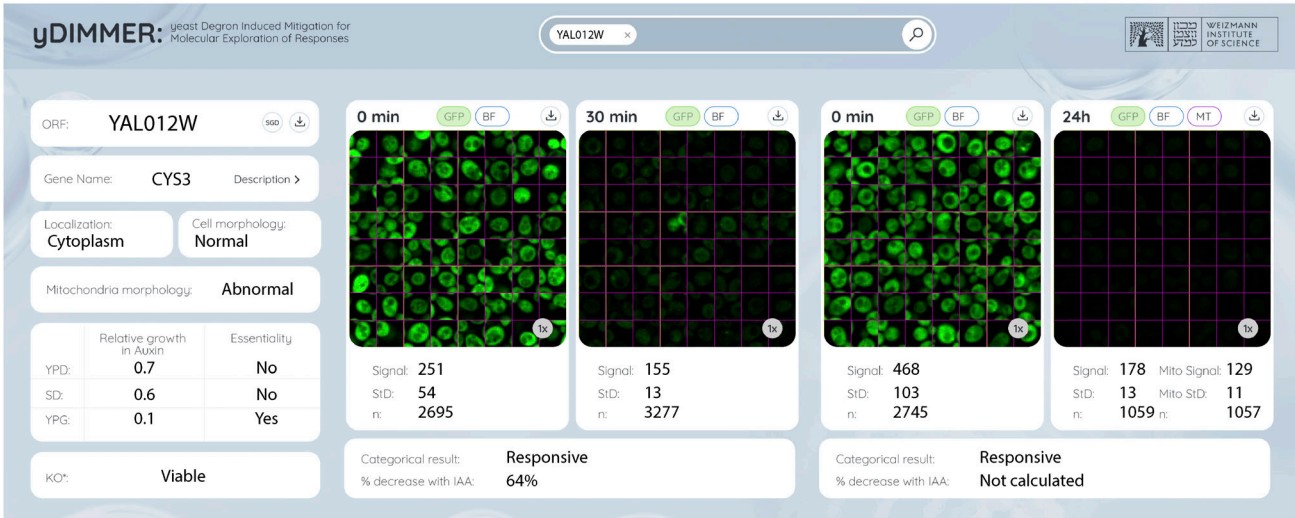

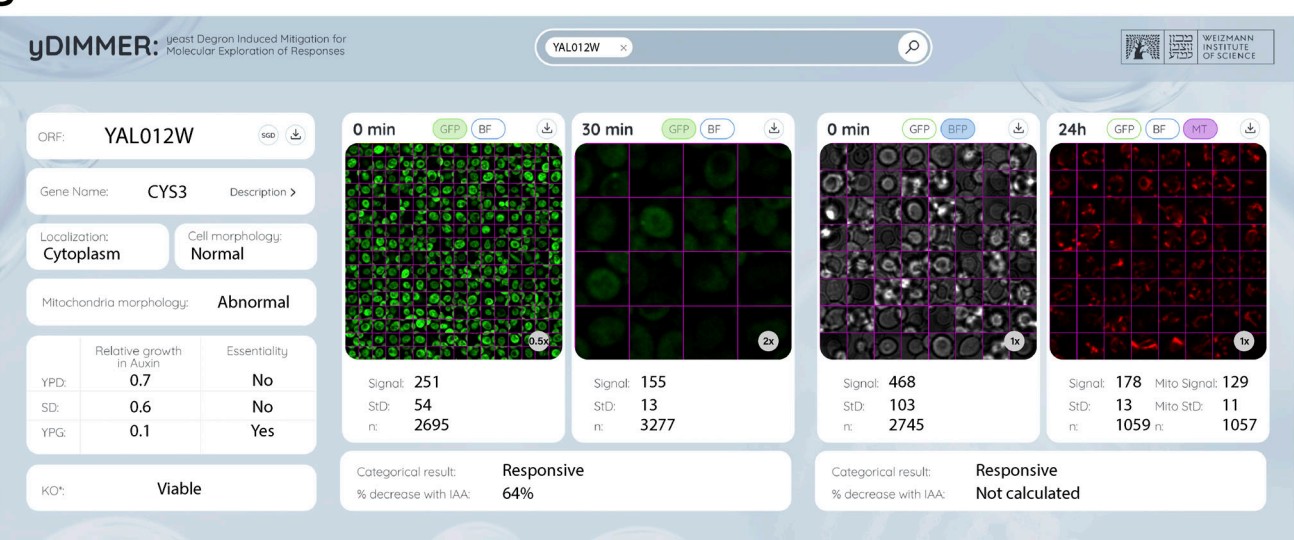

Figure S6.   **yDIMMER database allows access to the C' AID-GFP strains images and information. (A)** An example entry for the protein Cys3. Left: Information about the gene, including external resources as the link to the SGD (Engel et al., 2022; Wong et al., 2023), the GPF localization (Huh et al., 2003), and the YKO phenotype (Giaever et al., 2002); as well as the results from the morphology and essentiality analysis conducted with the C' AID-GFP library. Center: Fluorescent images from the short induction assay, together with their quantification. Right: Fluorescent images form the long induction assay, together with their quantification. Note that the long induction assay was performed under harsher illumination conditions so the quantification of the GFP signal will always be bigger than for the short assay at time 0 min. For the 24 h time point, there is information from the MitoTracker signal. **(B)** Same entry from the website, but this time displaying the zooming options (center) and the option to visualize the brightfield or the MitoTracker signal (for 24 h induction).

Provided online are Table S1, Table S2, Table S3, Table S4, Table S5, and Table S6. Table S1 lists all the strains present in the AID library and their features as used in the different analyses of this work. Table S2 contains all yeast strains used in this study. Table S3 contains all plasmids used in this study. Table S4 contains all primers used in this study, except the primers used for the library sequencing. Table S5 contains the oligos and primers used for the Anchor-seq protocol (library sequencing). Table S6 contains the oxygen consumption rates for the strains tested (Figs. 4 and S4).

