## [Peer Review File · The Journal of Cell Biology]

A proteome-wide yeast degron collection for the dynamic study of protein function

Rosario Valenti, Yotam David, Dunya Edilbi, Benjamin Dubreuil, Angela Boshnakovska, Yeynit Asraf, Tomer Meir-Salame, Ehud Sass, Peter Rehling, and Maya Schuldiner

Corresponding Author(s): Maya Schuldiner, Weizmann Institute of Science

Review Timeline:

Submission Date:	2024-09-19
Editorial Decision:	2024-10-30
Revision Received:	2024-11-06

Monitoring Editor: Jodi Nunnari

Scientific Editor: Dan Simon

Transaction Report:

DOI: <https://doi.org/10.1083/jcb.202409050>

Revision 0

Review #1

1. Evidence, reproducibility and clarity:

Evidence, reproducibility and clarity (Required)

****Summary****

This manuscript by the Schldiner group reports that the team has established a conditional degron library for 5110 yeast proteins. They utilized an improved version of an auxin-inducible degron (AID2) and fused the degron tag together with eGFP to the proteins' C-terminus (Fig. 1). Among about 3000 strains in which GFP expression was observed, 90% of the strains (2717 proteins) showed reduced expression or lethality upon the 5-Ph-IAA treatment for 24 h (Fig. 2). Mitochondrial proteins were more challenging to degrade because the ubiquitin-proteasome pathway does not operate within the organelle. The authors found that 347 strains showed a severe growth defect, and some showed this phenotype in a culture condition-dependent manner (Fig. 3). Moreover, the authors identified 11 new proteins affecting yeast growth, which had not been found in the YKO screening. Subsequently, the authors screened proteins whose loss affected mitochondrial distribution and morphology (MDM) and identified 220 previously undescribed candidates (Fig. 4). Furthermore, the authors revealed that the loss of proteins involved in the ergosterol biosynthesis pathway caused an MDM phenotype. Finally, the authors revisited the CDC screening and found additional CDC proteins which had not been identified in the original ts screening done by Lee Hartwell (Fig. 5).

****Major comments****

- Yeast strains in the library expressed an E3 subunit, OsTIR1(F74G), under the TEF2 promoter and were treated with 1 or 5 μ M 5-Ph-IAA on an agar plate or in a liquid culture, respectively. It is important to confirm that OsTIR1(F74G) and/or 5-Ph-IAA do not cause side effects in their experimental conditions. The authors did not confirm this point.
- Focusing on essential proteins for cell viability, the authors reported that 69% of essential proteins identified by the YKO screening showed a growth defect (Figure S3A, threshold <0.5). In other words, 31% (175 proteins) did not show a growth defect at all. Considering the fact that 90% of the strains showed target depletion, the depletion level was likely to be not enough for these proteins. The authors should discuss how the strain library can be improved to achieve better target depletion. Previous literature reported various possibilities, such as using a tandem degron tag and combining AID with the Tet promoter system (PMID 25181302, 26081484). Although optional, it would be wonderful if the authors would generate an improved library.

****Minor comments****

- 5-Ph-IAA is not auxin because it does not induce the auxin responses in plants (PMID 29355850). Therefore, the authors should be careful when they refer to 5-Ph-IAA and should not call it auxin.
- The authors should cite and discuss a previous paper showing an AID library for the yeast

essential genes (PMID 30670608).

- When they refer to Sld3, a reference is missing (Fig. 6, page 10). PMID11296242 is appropriate.

2. Significance:

Significance (Required)

This paper is technically robust and well-conducted. It presents a comprehensive study showcasing the effectiveness of the conditional degron library. Due to the rapid depletion of target proteins, cells exhibit immediate defects before an adaptive response arises. The described library and database will be invaluable for future screenings and studies across all aspects of yeast biology.

3. How much time do you estimate the authors will need to complete the suggested revisions:

Estimated time to Complete Revisions (Required)

(Decision Recommendation)

Between 1 and 3 months

4. Review Commons values the work of reviewers and encourages them to get credit for their work. Select 'Yes' below to register your reviewing activity at Web of Science Reviewer Recognition Service (formerly Publons); note that the content of your review will not be visible on Web of Science.

Yes

Review #2

1. Evidence, reproducibility and clarity:

Evidence, reproducibility and clarity (Required)

In this study, the authors developed a genome-wide library of conditional yeast mutants based on the AID2 technology. They carefully validated the library strains and characterized the responsiveness of the strains to the inducer 5-Ph-IAA. The library showed a high level of responsiveness in terms of protein level reduction based on imaging analysis. Moreover, the authors performed phenotypic analyses, including examining growth phenotypes in three different media, mitochondrial staining intensity and patterns, and cell cycle arrest phenotype. The results of these analyses demonstrate the utility of this library.

I only have a few minor comments and suggestions.

Point 1

Introduction

"Partially, this can be attributed to limitations in existing tools, such as compensatory mutations arising after gene deletion (Hughes et al. 2000; Teng et al. 2013) that would mask informative phenotypes, and short-term cellular rewiring, that obscures the correct coupling between perturbation and observed phenotype."

It is unclear to me what "short-term cellular rewiring" means. I suggest the authors consider providing examples and adding citations to explain this concept.

Point 2

Introduction

"Perturbing the translation of a gene, or even the levels of its mRNA (Schuldiner et al. 2005), might be too slow to report on an immediate phenotype, as seen when contrasting phenotypes of the heat-inducible-degron (Kanemaki et al. 2003) with literature reports on phenotypes visualized by other methods (Yu et al. 2006)."

In Yu et al. 2006, I could not find any descriptions of discrepancies between Kanemaki et al. 2003 and Yu et al. 2006. It would be helpful to provide an example here.

Point 3

Results

"Furthermore, many conditional depletion systems have shown to be more efficient with highly expressed genes (Arita et al. 2021). We noted our system follows the opposite trend (Supplementary Figure 3B), making it highly complementary to other approaches."

What could be the reason that highly expressed genes are less likely to show a growth phenotype? Perhaps the authors can add some possible explanations.

Point 4

Results

"or Erg11, where the depletion is slower probably due to its C' facing the lumen of the ER (Supplementary Figure 4B and C)."

To my knowledge, the C-terminus of Erg11 faces the cytosol. See Figure 1 of Monk et al. 2014 (PMID: 24613931).

Point 5

Figure legend

Figure 5

"D) Flow cytometry measurements for strains in A, with or without a 3h induction."

I think "strains in A" should be "strains in B".

2. Significance:

Significance (Required)

The conditional mutant strain library generated in this study is a significant and valuable addition to the functional genomic tools available for budding yeast. Furthermore, the authors will ensure easy access to the strain images and phenotype information through a user-friendly website. Overall, both the strain library and the data generated in this study will serve as valuable resources.

3. How much time do you estimate the authors will need to complete the suggested revisions:

Estimated time to Complete Revisions (Required)

(Decision Recommendation)

Less than 1 month

4. Review Commons values the work of reviewers and encourages them to get credit for their work. Select 'Yes' below to register your reviewing activity at Web of Science Reviewer Recognition Service (formerly Publons); note that the content of your review will not be visible on Web of Science.

Yes

Review #3

1. Evidence, reproducibility and clarity:

Evidence, reproducibility and clarity (Required)

Summary: In this manuscript, the authors present the construction and analysis of an *S. cerevisiae* library in which most genes are fused to an auxin-inducible degron and gfp. They demonstrate the usefulness of this library by screening for genes required for normal mitochondrial function and for cell division, and, in both cases, making new discoveries. The manuscript is clearly written and the results and the library will be of great interest to the yeast community. Minor comments are below.

1. Since we don't know the sensitivity of the methods used for protein detection, it is likely misleading in some cases to use the term "complete degradation." Please substitute "degradation to undetectable levels" in place of "complete degradation" throughout the manuscript.
2. Figure 1C - For the Westerns, in place of "POI" please put the protein name. And please put tick marks for the MW markers.
3. Figure 1C - The Western for Tub3 is cropped too close to the band, especially since the gels smiling a bit. Please recrop to show more of the blot above the Tub3 band.
4. Figure 1D - Were the values for the Western blots also normalized to Act1 levels for each time

point? Please clarify.

5. Supplementary Figure 1C and 1D - Please provide a little more detail on what was done here. The way that it is presented seems in reverse order. First, 90% were recovered (currently D), then of those 90%, all but 12 were correct (currently C).

6. Figure 2 and Supplementary Figure 2 - It is not possible to make conclusions for the kinetics of depletion for many of the proteins using time points at 30 minutes and 24 hours. For example, Gpp1 (in Figure 1) is not depleted at 30 minutes, but it is mostly depleted by 90 minutes. Therefore, the authors should be more cautious in how they term the class of proteins that are not depleted by 30 minutes.

7. Figure 3B legend - Please fix the legend - "but around 500"???

2. Significance:

Significance (Required)

This library will be useful to the yeast community.

3. How much time do you estimate the authors will need to complete the suggested revisions:

Estimated time to Complete Revisions (Required)

(Decision Recommendation)

Less than 1 month

Yes

Point by point response to reviewers' suggestions.

Reviewer 1.

Major comments

- Yeast strains in the library expressed an E3 subunit, OsTIR1(F74G), under the TEF2 promoter and were treated with 1 or 5 μ M 5-Ph-IAA on an agar plate or in a liquid culture, respectively. It is important to confirm that OsTIR1(F74G) and/or 5-Ph-IAA do not cause side effects in their experimental conditions. The authors did not confirm this point.

We thank the reviewer for prompting us to do this important control. After conducting growth assays in liquid media, we indeed saw a small growth defect on the strains containing the OsTIR1(F74G) when compared to their parental strain (8.3% see plot below). We did not see a similar effect from inserting only the resistance marker, or from expressing a GFP protein with a TEF2 promoter integrated into the same locus (data not shown). Since the growth defect is small, and in the library the strains are always compared to themselves, but without 5-Ph-IAA induction, we expect the results from using the AID-GFP to be robust. However, we have now clearly stated this point in the manuscript.

We did not observe any growth effect from the 5-Ph-IAA treatment, neither in the strains containing the OsTIR1(F74G) or their parental strains.

Impact of the AID system on growth. A) Growth curves of control (WT) strain (blue, BY4741), the same strain transformed with the OsTIR1(F74G) and resistance to nourseothricin (NAT, green), or WT transformed only with the resistance to NAT (orange); all with 5 μ M 5-Ph-IAA in DMSO (dark colors) or 5 μ M DMSO only (light colors). The strain containing the OsTIR1(F74G) presents a small growth defect when compared to WT or to the strain with matching resistance. **B)** Bar graph displaying the duplication time calculated from result on A. From a 2-way ANOVA, the strain expressing the OsTIR1(F74G) resulted significantly different to the WT strain (p -value 0.0333, difference $8.3 \pm 0.1\%$). Treatment with 5-Ph-IAA did not produce any effect.

- Focusing on essential proteins for cell viability, the authors reported that 69% of essential proteins identified by the YKO screening showed a growth defect (Figure S3A, threshold <0.5). In other words, 31% (175 proteins) did not show a growth defect at all. Considering the fact that 90% of the strains showed target depletion, the depletion level was likely to be

not enough for these proteins. The authors should discuss how the strain library can be improved to achieve better target depletion. Previous literature reported various possibilities, such as using a tandem degron tag and combining AID with the Tet promoter system (PMID 25181302, 26081484). Although optional, it would be wonderful if the authors would generate an improved library.

The reviewer's concern on this point is valid. When considering reasons why 175 strains reported as essential by the YKO do not show a growth defect in the AID library, we see four options. First, there is the possibility that original reports could be inaccurate. Mining the literature for such cases, we found 21 cases where consecutive low throughput studies were able to successfully delete a gene listed as essential in the YKO (*ARP9*, *BIG1*, *CDC36*, *FHL1*, *FOL3*, *LAS17*, *MMF1*, *PTR3*, *ROT1*, *RRP14*, *SDH3*, *SSY1*, *SSY5*, *TOM20*, *TRM5*, *UTR5*, *YJR012C*). From those, we are in agreement with 17 as they do not display a growth defect in the present AID collection.

Second, in our study the essentiality was measured in a high throughput manner as a continuous variable. We then selected the threshold of <0.5 relative growth to define a strain as presenting a "severe growth defect". As illustrated in Supplementary Figure 3, this threshold captures 385 strains listed in the YKO as essential, while it excludes 175 strains. Of those 175 strains with no severe growth defect, 59 had a moderate growth defect (relative growth between 0.5 and 0.86) and 116 had no growth defect (relative growth >0.86), see figure below. It remains possible that in the long term, a moderate growth defect would be enough to cause the demise of the strain. Furthermore, of the 175 strains in contention, 15 showed less than 0.5 relative growth in minimal media, 13 only in respiration media, and 10 in both. This discrepancy highlights an inescapable level of noise when measuring the ratios that might lead to some ambiguity in the designation of several strains.

Comparison of growth defects in the AID library with the essentiality determination of the YKO collection. The relative growth calculated from the colony sizes in YPD agar with 5 μ M 5-Ph-IAA vs only YPD is plotted for all the verified responsive strains of the AID-GFP collection. The strains carrying the tag in a gene defined as essential by the YKO are marked in blue while all the other categories (“competitive fitness decreased”, “viable” and “absent”) are grouped under “other” and colored yellow. Pie charts show the distribution of AID-GFP phenotypes in the YKO categories (top), and of YKO phenotypes in the growth defect categories (right). The threshold of <0.5 relative growth captures 385 strains deemed essential by the YKO. Another 59 strains present a moderate growth phenotype.

Third, it could be that our degradation strategy did not enable enough depletion to unmask the essentiality of the protein. To address this case, we conducted the analysis plotted below. We separated the strains by abundance and phenotype, and we evaluated the median fluorescence for each strain after induction of the system. If in a category, the depletion is incomplete, their “residual fluorescence” should be higher. After performing a 2-way ANOVA, both factors (abundance and phenotype) were significant. We performed a Tukey's multiple comparisons test for the phenotype factor and we saw that the only different group was that with the inviable strains with a severe growth defect (vs Inviable with no severe growth defect, p-value 0.0054; vs viable <0.0001). This doesn't readily support the expectations for the case where incomplete degradation prevents the growth defect. On the contrary, it points at strains showing a defect having a higher residual fluorescence. We speculate this enhanced residual fluorescence might come from an incomplete degradation of the tagged protein after the cell demise, or from an increased autofluorescence. In our hands many dead cells present increased autofluorescence, as evident by evaluating the signal across several channels. **This result does not support the hypothesis of an incomplete degradation being a strong determinant on whether a strain would display a growth defect or not.**

Residual GFP fluorescence for each strain. Box plot displaying the median GFP fluorescence calculated for each responsive strain after 24h of induction, separated by mRNA abundance (with

Quintile 1 as the most abundant), and phenotype (inviable refers to the YKO classification; with or without growth defect refers to the AID-GFP collection). The only group with elevated residual GFP fluorescence is the one displaying a growth defect (2-way ANOVA, vs Inviable with no severe growth defect, p-value 0.0054; vs viable <0.0001). Higher values might indicate an incomplete degradation; alternatively, they might arise from an increase in autofluorescence.

Finally, it could be that the essentiality of some of the discrepant cases is condition dependent, and our conditions as well as genetic backgrounds are slightly different than those used in the YKO that revealed it. As the authors of the YKO collection pointed out, “[...] essential genes are better described as a spectrum rather than a binary distinction” (Giaever and Nislow 2014). Indeed, over the years, numerous studies support the notion that essentiality is both strain-specific and/or condition-specific (for just a small example: Li et al. 2019; Sadhu et al. 2018; Van Leeuwen et al. 2016). Our library strain is indeed made in a different genetic background than the YKO collection. As an example, the YKO strain is deficient in leucine biosynthesis (*len2Δ0*) and three of the genes not showing a growth defect in our collection were found to be non-essential when the strain is prototrophic for leucine as in our strain (Sadhu et al. 2018).

Taken together, these analyses suggest that a combination of factors, both biological and technical, are behind the discrepancies observed with the YKO. We therefore hypothesize that a new library would not necessarily address the lack of growth defect in most cases.

Regarding the creation of a library with both translation and transcriptional repression, this very interesting approach could address some of the limitations of our tool. However, our library was generated from a C-SWAT library (Meurer et al. 2018), therefore the high throughput modification of the 5' of each gene is not accessible. Instead, this approach would be applicable for an N-SWAT library, for those proteins that do not require a signal peptide or MTS (Weill et al. 2018; Yofe et al. 2016) but this would be an entirely new endeavor and outside the scope of this work. It would also be a great way to improve specific strains of interest in low throughput studies. We have now referred to this in the discussion.

Minor comments

- 5-Ph-IAA is not auxin because it does not induce the auxin responses in plants (PMID 29355850). Therefore, the authors should be careful when they refer to 5-Ph-IAA and should not call it auxin.

Thank you for noticing this, we have now fixed this error.

- The authors should cite and discuss a previous paper showing an AID library for the yeast essential genes (PMID 30670608).

Thank you for noticing this major omission. We of course intended to cite it, and this was lost during subsequent rewriting. We have now re-incorporated the citation into the text.

- When they refer to Sld3, a reference is missing (Fig. 6, page 10). PMID11296242 is appropriate.

Apologies. It has now been incorporated, thank you.

Reviewer 2.

Point 1: Introduction

"Partially, this can be attributed to limitations in existing tools, such as compensatory mutations arising after gene deletion (Hughes et al. 2000; Teng et al. 2013) that would mask informative phenotypes, and short-term cellular rewiring, that obscures the correct coupling between perturbation and observed phenotype."

It is unclear to me what "short-term cellular rewiring" means. I suggest the authors consider providing examples and adding citations to explain this concept.

We consider "short-term cellular rewiring" as transcriptional, post-transcriptional or metabolic compensatory mechanisms that mask the effects of genetic alterations, such that the resulting phenotype is no longer informative. For example, genes tend to be upregulated upon deletion of their functionally similar paralogs (Kafri, Bar-Even, and Pilpel 2005). In other instances, the activation of a transcriptional stress response following depletion of a single protein can rescue its levels and obscure the emergence of a phenotype (Schuldiner et al. 2005).

We expect our approach to allow a window of opportunity for the study of the cellular response before there is such a compensatory response. These short-lived dynamic states could be particularly informative in the study of metabolism, which is rapidly equilibrated. We have added a longer discussion as well as the two examples above to the introduction of the manuscript.

Point 2: Introduction

"Perturbing the translation of a gene, or even the levels of its mRNA (Schuldiner et al. 2005), might be too slow to report on an immediate phenotype, as seen when contrasting phenotypes of the heat-inducible-degron (Kanemaki et al. 2003) with literature reports on phenotypes visualized by other methods (Yu et al. 2006)."

In Yu et al. 2006, I could not find any descriptions of discrepancies between Kanemaki et al. 2003 and Yu et al. 2006. It would be helpful to provide an example here.

The wording of this sentence is indeed confusing. In both Kanemaki et al. 2003 and Yu et al. 2006 the *TET*-off strain of *MCM4* or *CDC54* (*YPR019W*) arrests with 2C DNA content after treatment. The contrast is between using a transcriptional repression or a fast protein depletion. The wording has been changed to:

"One example (Kanemaki et al. 2003), demonstrates that for the essential helicase *MCM4*, both a rapid protein destabilization and the repression of transcription were lethal, but the different rates of protein depletion in each method resulted in cells being arrested in different stages of the cell cycle."

Point 3: Results

"Furthermore, many conditional depletion systems have shown to be more efficient with highly expressed genes (Arita et al. 2021). We noted our system follows the opposite trend (Supplementary Figure 3B), making it highly complementary to other approaches."

What could be the reason that highly expressed genes are less likely to show a growth phenotype? Perhaps the authors can add some possible explanations.

There could be multiple reasons that could generate this trend. One option is that highly abundant mRNAs might allow for a higher translation rate, replenishing the depleted protein. A second possible explanation could be the saturation of the degradation machinery derived from degrading such abundant proteins. A third option might be that the regulation of abundance is tighter for these higher expressed proteins, and that their reduction triggers its translational

induction. Finally, a fourth possibility would be that, despite their high levels, they are only required in levels below our detection threshold, and that their abundance is a functional buffer.

In the first three cases, we would expect a higher fluorescent signal after induction. When conducting the analysis in response to the first reviewer's comment, we encountered that strains from the most abundant Quintile (1) indeed had a significantly increased fluorescence after induction (2-way ANOVA, $p < 0.0001$). This elevated residual GFP fluorescence was still present when removing strains displaying a severe growth defect (whose fluorescence might be not reliable to analyze). We therefore can speculate that the depletion of highly abundant proteins might not be efficient enough to trigger a growth defect. However, which of the three reasons stated before is behind this effect is an interesting point, beyond the scope of our manuscript. Regardless of the reason, we have now clearly stated the point of incomplete depletion for highly abundant proteins in the discussion.

Point 4: Results

"or Erg11, where the depletion is slower probably due to its C' facing the lumen of the ER (Supplementary Figure 4B and C)."

To my knowledge, the C-terminus of Erg11 faces the cytosol. See Figure 1 of Monk et al. 2014 (PMID: 24613931).

Thank you for bringing our attention to this work. In light of it, we have removed the sentence where we speculated that the slow degradation of Erg11-AID-GFP was due to its C' facing the lumen of the ER.

Point 5: Figure legend Figure 5

"D) Flow cytometry measurements for strains in A, with or without a 3h induction."

I think "strains in A" should be "strains in B".

Thank you for noticing this, we have now fixed this error.

Reviewer 3.

1. Since we don't know the sensitivity of the methods used for protein detection, it is likely misleading in some cases to use the term "complete degradation." Please substitute "degradation to undetectable levels" in place of "complete degradation" throughout the manuscript.

We completely agree with this point. In the revised version the phrasing has been corrected in accordance with this.

2. Figure 1C - For the Westerns, in place of "POI" please put the protein name. And please put tick marks for the MW markers.

We have incorporated these suggestions.

3. Figure 1C - The Western for Tub3 is cropped too close to the band, especially since the gels smiling a bit. Please recrop to show more of the blot above the Tub3 band.

We have re-cropped the image as suggested.

4. Figure 1D - Were the values for the Western blots also normalized to Act1 levels for each

time point? Please clarify.

Yes, the protein levels were normalized by the Act1 levels measured in each lane. The method section has been updated to reflect the whole image quantification process applied to the membranes.

5. Supplementary Figure 1C and 1D - Please provide a little more detail on what was done here. The way that it is presented seems in reverse order. First, 90% were recovered (currently D), then of those 90%, all but 12 were correct (currently C).

We thank you for the suggestion to flip the figures, it does make for a more logical flow.

Originally, in D, 3670 colonies were pooled, their DNA extracted, enriched for the AID modified region with the Anchor-seq method, and sequenced. From those, 90% of the genes were detected (recovered) by sequencing. Of those 3308 genes whose sequence was recovered, only 12 displayed issues with the tag.

Now we present the coverage as C and the proportion of functional tags as D, and we have also clarified the methodology and rationale of the process. Furthermore, we have re-sequenced the entire library, since our original run had technical issues which prevented the submission of nearly one fourth of the collection. The updated values are displayed (5084/5170 recovered, 5063/5084 with the correct tag).

6. Figure 2 and Supplementary Figure 2 - It is not possible to make conclusions for the kinetics of depletion for many of the proteins using time points at 30 minutes and 24 hours. For example, Gpp1 (in Figure 1) is not depleted at 30 minutes, but it is mostly depleted by 90 minutes. Therefore, the authors should be more cautious in how they term the class of proteins that are not depleted by 30 minutes.

Thank you for noticing it. We have fixed the wording.

7. Figure 3B legend - Please fix the legend - "but around 500"???

We have clarified the writing

Bibliography:

- Kafri, Ran, Arren Bar-Even, and Yitzhak Pilpel. 2005. “Transcription Control Reprogramming in Genetic Backup Circuits.” *Nature Genetics* 37(3):295–99. doi: 10.1038/ng1523.
- Kanemaki, Masato, Alberto Sanchez-Diaz, Agnieszka Gambus, and Karim Labib. 2003. “Functional Proteomic Identification of DNA Replication Proteins by Induced Proteolysis in Vivo.” *Nature* 423(6941):720–25. doi: 10.1038/nature01692.
- Li, Jun, Hai-Tao Wang, Wei-Tao Wang, Xiao-Ran Zhang, Fang Suo, Jing-Yi Ren, Ying Bi, Ying-Xi Xue, Wen Hu, Meng-Qiu Dong, and Li-Lin Du. 2019. “Systematic Analysis Reveals the Prevalence and Principles of Bypassable Gene Essentiality.” *Nature Communications* 10(1):1002. doi: 10.1038/s41467-019-08928-1.
- Meurer, Matthias, Yuanqiang Duan, Ehud Sass, Ilia Kats, Konrad Herbst, Benjamin C. Buchmuller, Verena Dederer, Florian Huber, Daniel Kirrmaier, Martin Štefl, Koen Van Laer, Tobias P. Dick, Marius K. Lemberg, Anton Khmelinskii, Emmanuel D. Levy, and Michael Knop. 2018. “Genome-Wide C-SWAT Library for High-Throughput Yeast Genome Tagging.” *Nature Methods* 15(8):598–600. doi: 10.1038/s41592-018-0045-8.
- Sadhu, Meru J., Joshua S. Bloom, Laura Day, Jake J. Siegel, Sriram Kosuri, and Leonid Kruglyak. 2018. “Highly Parallel Genome Variant Engineering with CRISPR–Cas9.” *Nature Genetics* 50(4):510–14. doi: 10.1038/s41588-018-0087-y.
- Van Leeuwen, Jolanda, Carles Pons, Joseph C. Mellor, Takafumi N. Yamaguchi, Helena Friesen, John Koschwanez, Mojca Mattiazzi Ušaj, Maria Pechlaner, Mehmet Takar, Matej Ušaj, Benjamin VanderSluis, Kerry Andrusiak, Pritpal Bansal, Anastasia Baryshnikova, Claire E. Boone, Jessica Cao, Atina Cote, Marinella Gebbia, Gene Horecka, Ira Horecka, Elena Kuzmin, Nicole Legro, Wendy Liang, Natascha Van Lieshout, Margaret McNee, Bryan-Joseph San Luis, Fatemeh Shaeri, Ermira Shuteriqi, Song Sun, Lu Yang, Ji-Young Youn, Michael Yuen, Michael Costanzo, Anne-Claude Gingras, Patrick Aloy, Chris Oostenbrink, Andrew Murray, Todd R. Graham, Chad L. Myers, Brenda J. Andrews, Frederick P. Roth, and Charles Boone. 2016. “Exploring Genetic Suppression Interactions on a Global Scale.” *Science* 354(6312):aag0839. doi: 10.1126/science.aag0839.
- Weill, Uri, Ido Yofe, Ehud Sass, Bram Stynen, Dan Davidi, Janani Natarajan, Reut Ben-Menachem, Zohar Avihou, Omer Goldman, Nofar Harpaz, Silvia Chuartzman, Kiril Kniazev, Barbara Knoblach, Janina Laborenz, Felix Boos, Jacqueline Kowarzyk, Shifra Ben-Dor, Einat Zalckvar, Johannes M. Herrmann, Richard A. Rachubinski, Ophry Pines, Doron Rapaport, Stephen W. Michnick, Emmanuel D. Levy, and Maya Schuldiner. 2018. “Genome-Wide SWAp-Tag Yeast Libraries for Proteome Exploration.” *Nature Methods* 15(8):617–22. doi: 10.1038/s41592-018-0044-9.
- Yofe, Ido, Uri Weill, Matthias Meurer, Silvia Chuartzman, Einat Zalckvar, Omer Goldman, Shifra Ben-Dor, Conny Schütze, Nils Wiedemann, Michael Knop, Anton Khmelinskii, and Maya Schuldiner. 2016. “One Library to Make Them All: Streamlining the Creation of Yeast Libraries via a SWAp-Tag Strategy.” *Nature Methods* 13(4):371–78. doi: 10.1038/nmeth.3795.

October 30, 2024

RE: JCB Manuscript #202409050T

Prof. Maya Schuldiner
Weizmann Institute of Science
Department of Molecular Genetics
Meyer Bldg
Rehovot 7610001
Israel

Dear Prof. Schuldiner,

Thank you for submitting your revised manuscript entitled "A proteome-wide yeast degron collection for the dynamic study of protein function." We would be happy to publish your paper in JCB pending final revisions necessary to meet our formatting guidelines (see details below). Please also address Reviewer #1's request to incorporate data from the rebuttal document into the manuscript.

A. MANUSCRIPT ORGANIZATION AND FORMATTING:

1) Text limits: Character count for Tools is < 40,000, not including spaces. Count includes title page, abstract, introduction, results, discussion, and acknowledgments. Count does not include materials and methods, figure legends, references, tables, or supplemental legends.

2) Figure formatting: Tools may have up to 10 main text figures. Scale bars must be present on all microscopy images, including inset magnifications. Molecular weight or nucleic acid size markers must be included on all gel electrophoresis.

Also, please avoid pairing red and green for images and graphs to ensure legibility for color-blind readers. If red and green are paired for images, please ensure that the particular red and green hues used in micrographs are distinctive with any of the colorblind types. If not, please modify colors accordingly or provide separate images of the individual channels.

3) Statistical analysis: Error bars on graphic representations of numerical data must be clearly described in the figure legend. The number of independent data points (n) represented in a graph must be indicated in the legend. Please, indicate whether 'n' refers to technical or biological replicates (i.e. number of analyzed cells, samples or animals, number of independent experiments). If independent experiments with multiple biological replicates have been performed, we recommend using distribution-reproducibility SuperPlots (please see Lord et al., JCB 2020) to better display the distribution of the entire dataset, and report statistics (such as means, error bars, and P values) that address the reproducibility of the findings.

Statistical methods should be explained in full in the materials and methods. For figures presenting pooled data the statistical measure should be defined in the figure legends. Please also be sure to indicate the statistical tests used in each of your experiments (both in the figure legend itself and in a separate methods section) as well as the parameters of the test (for example, if you ran a t-test, please indicate if it was one- or two-sided, etc.). Also, if you used parametric tests, please indicate if the data distribution was tested for normality (and if so, how). If not, you must state something to the effect that "Data distribution was assumed to be normal but this was not formally tested."

4) Materials and methods: Should be comprehensive and not simply reference a previous publication for details on how an experiment was performed. Please provide full descriptions (at least in brief) in the text for readers who may not have access to referenced manuscripts. The text should not refer to methods "...as previously described."

5) For all cell lines, vectors, constructs/cDNAs, etc. - all genetic material: please include database / vendor ID (e.g., Addgene, ATCC, etc.) or if unavailable, please briefly describe their basic genetic features, even if described in other published work or gifted to you by other investigators (and provide references where appropriate). Please be sure to provide the sequences for all of your oligos: primers, si/shRNA, RNAi, gRNAs, etc. in the materials and methods. You must also indicate in the methods the source, species, and catalog numbers/vendor identifiers (where appropriate) for all of your antibodies, including secondary. If antibodies are not commercial, please add a reference citation if possible.

6) Microscope image acquisition: The following information must be provided about the acquisition and processing of images:

- Make and model of microscope
- Type, magnification, and numerical aperture of the objective lenses
- Temperature
- Imaging medium
- Fluorochromes
- Camera make and model
- Acquisition software
- Any software used for image processing subsequent to data acquisition. Please include details and types of operations involved (e.g., type of deconvolution, 3D reconstitutions, surface or volume rendering, gamma adjustments, etc.).

7) References: There is no limit to the number of references cited in a manuscript. References should be cited parenthetically in the text by author and year of publication. Abbreviate the names of journals according to PubMed.

8) Supplemental materials: Tools generally may have up to 5 supplemental figures and 10 videos. You currently exceed this limit but, in this case, we will be able to give you the extra space. Additionally, figures cannot span multiple pages as your current Supplemental Figure 4 does. Please reorganize this figure and split up the data as necessary into other or new figures. Please also note that tables, like figures, should be provided as individual, editable files. A summary of all supplemental material should appear at the end of the Materials and methods section. Please include one brief sentence per item.

9) eTOC summary: A ~40-50 word summary that describes the context and significance of the findings for a general readership should be included on the title page. The statement should be written in the present tense and refer to the work in the third person. It should begin with "First author name(s) et al..." to match our preferred style.

10) Conflict of interest statement: JCB requires inclusion of a statement in the acknowledgements regarding competing financial interests. If no competing financial interests exist, please include the following statement: "The authors declare no competing financial interests." If competing interests are declared, please follow your statement of these competing interests with the following statement: "The authors declare no further competing financial interests."

11) A separate author contribution section is required following the Acknowledgments in all research manuscripts. All authors should be mentioned and designated by their first and middle initials and full surnames. We encourage use of the CRediT nomenclature (<https://casrai.org/credit/>).

12) ORCID IDs: ORCID IDs are unique identifiers allowing researchers to create a record of their various scholarly contributions in a single place. Please note that ORCID IDs are required for all authors. At resubmission of your final files, please be sure to provide your ORCID ID and those of all co-authors.

13) JCB requires authors to submit Source Data used to generate figures containing gels and Western blots with all revised manuscripts. This Source Data consists of fully uncropped and unprocessed images for each gel/blot displayed in the main and supplemental figures. Since your paper includes cropped gel and/or blot images, please be sure to provide one Source Data file for each figure that contains gels and/or blots along with your revised manuscript files. File names for Source Data figures should be alphanumeric without any spaces or special characters (i.e., SourceDataF#, where F# refers to the associated main figure number or SourceDataFS# for those associated with Supplementary figures). The lanes of the gels/blots should be labeled as they are in the associated figure, the place where cropping was applied should be marked (with a box), and molecular weight/size standards should be labeled wherever possible. Source Data files will be directly linked to specific figures in the published article.

14) Journal of Cell Biology now requires a data availability statement for all research article submissions. These statements will be published in the article directly above the Acknowledgments. The statement should address all data underlying the research presented in the manuscript. Please visit the JCB instructions for authors for guidelines and examples of statements at (<https://rupress.org/jcb/pages/editorial-policies#data-availability-statement>).

B. FINAL FILES:

-- High-resolution figure and MP4 video files: See our detailed guidelines for preparing your production-ready images,

<https://jcb.rupress.org/fig-vid-guidelines>.

Thank you for your attention to these final processing requirements. Please revise and format the manuscript and upload materials within 7 days. If you need an extension for whatever reason, please let us know and we can work with you to determine a suitable revision period.

Thank you for this interesting contribution, we look forward to publishing your paper in Journal of Cell Biology.

Sincerely,

Jodi Nunnari, PhD
Editor-in-Chief
Journal of Cell Biology

Dan Simon, PhD
Scientific Editor
Journal of Cell Biology

Reviewer #1 (Comments to the Authors (Required)):

I am glad to read the revised manuscript and the response letter, which have cleared my concerns.

As the authors presented in the response letter ("impact of the AID system on growth"), they found that the yeast strain expressing OstTIR1(F74G) showed slightly slower growth than WT, and the treatment with 5 μ M 5-Ph-IAA did not impact cell proliferation. I understand the authors used yeast strains expressing OstTIR1(F74G) in all experiments and compared their phenotypes with or without 5-Ph-IAA, indicating that the observed defects were unrelated to OstTIR1(F74G) expression. However, it is important to disclose these findings so that readers can evaluate the data more precisely. I highly suggest that the data presented in "Impact of the AID system on growth" should be incorporated into the final version of the manuscript.

Reviewer #2 (Comments to the Authors (Required)):

The revision has fully addressed all my concerns. I support the publication of this paper.

Reviewer #3 (Comments to the Authors (Required)):

The authors have done an excellent job of addressing the reviewers' comments. I think that the paper is now acceptable for publication.

Prof. Maya Schuldiner

Incumbent of Omenn and Darling Chair in Molecular Genetics
Department of Molecular Genetics, Weizmann Institute of Science
Webpage: <http://www.weizmann.ac.il/molgen/Maya/>

Manuscript number: 202409050TR

Corresponding author(s): Maya, Schuldiner

November 6th 2024

Dear Editors,

We thank reviewer 1 for suggesting adding the data "impact of the AID system on growth" to the manuscript body. We have now done so, and this is reported in the new supplementary figure 1A and B.

We thank you and all the reviewers for their comments, their time and work in reviewing this manuscript.

Sincerely yours,

Maya Schuldiner.